# Dynamic landscape of the intracellular termini of acid-sensing ion channel 1a

**Megan M Cullinan[1], Robert C Klipp[1], Abigail Camenisch[2], John R Bankston[1]\***

[1]Department of Physiology and Biophysics, University of Colorado Anschutz Medical Campus, Aurora, United States; [2]University of Arizona College of Medicine, Tuscon, United States

**Abstract** Acid-sensing ion channels (ASICs) are trimeric proton-gated sodium channels. Recent work has shown that these channels play a role in necroptosis following prolonged acidic exposure like occurs in stroke. The C-terminus of ASIC1a is thought to mediate necroptotic cell death through interaction with receptor interacting serine threonine kinase 1 (RIPK1). This interaction is hypothesized to be inhibited at rest via an interaction between the C- and N-termini which blocks the RIPK1 binding site. Here, we use two transition metal ion FRET methods to investigate the conformational dynamics of the termini at neutral and acidic pH. We do not find evidence that the termini are close enough to be bound while the channel is at rest and find that the termini may modestly move closer together during acidification. At rest, the N-terminus adopts a conformation parallel to the membrane about 10 Å away. The distal end of the C-terminus may also spend time close to the membrane at rest. After acidification, the proximal portion of the N-terminus moves marginally closer to the membrane whereas the distal portion of the C-terminus swings away from the membrane. Together these data suggest that a new hypothesis for RIPK1 binding during stroke is needed.

**\*For correspondence:**
john.bankston@cuanschutz.edu

**Competing interest:** The authors declare that no competing interests exist.

## eLife assessment

This **valuable** study illuminates molecular movements of acid-sensing ion channels by combining advanced chemical biology and biophysical techniques. The evidence for the main claim, lack of interaction of molecular termini, is **compelling** and challenges prior models. This work is expected to pique interest in the ion channel signaling field, providing a fresh perspective.

## Introduction

Acid-sensing ion channels (ASICs) are members of the Deg/ENaC (Degenerin/Epithelial Sodium Channel) superfamily, with relatives including the mechanosensitive MEC4 from *C. elegans,* Pickpocket from *D. melanogaster* and most recently identified, the heat-activated BRNTaC1 from *S.s mutalns* (*Yao et al., 2023*). ASICs are trimeric, proton activated, sodium selective channels located primarily in the central and peripheral nervous systems. In mammals, there are four ASIC genes giving rise to six unique isoforms that can assemble into homo- and heteromeric channels. ASICs have been shown to be involved in diverse physiological and pathophysiological conditions including pain sensing (*Bässler et al., 2001*; *Mazzuca et al., 2007*; *Sluka et al., 2009*), mechanotransduction (*Hermanstyne et al., 2008*), fear conditioning (*Coryell et al., 2008*; *Du et al., 2014*), and memory and learning (*Wemmie et al., 2002*).

The first structure of a member of this family, ASIC1 from chicken (cASIC1), was solved at 1.9 Å and revealed the overall architecture of the channel (*Jasti et al., 2007*). A single subunit of cASIC1 is comprised of a large cysteine-rich extracellular domain, two transmembrane domains and short

intracellular N- and C-termini. More recent structures of cASIC1 from the same group using CryoEM on channels solubilized with styrene-maleic acid copolymer revealed that residues 19–41 reenter the membrane and form part of the lower permeation path: a feature that was termed the re-entrant loop (*Yoder and Gouaux, 2020*). Nonetheless, the C-terminus and the cytosolic portion of the N-terminus remain unresolved. It is likely these domains are highly dynamic and largely unstructured.

Despite the lack of structure, the termini play a critical role in channel function and regulation. ASICs have been shown to interact with a wide variety of intracellular proteins (*Cullinan et al., 2021*) like stomatin (*Brand et al., 2012*; *Klipp et al., 2020*; *Price et al., 2004*), NHERF (*Deval et al., 2006*), and CIPP (*Anzai et al., 2002*). These proteins can influence trafficking, gating, and localization. Moreover, mutations and truncations in the termini have been shown to impact channel gating as well (*Li et al., 2021*).

Of critical physiological importance, ASIC1a has a clearly demonstrated role in cell death resulting from ischemic stroke (*Gao et al., 2005*; *Wang et al., 2020*; *Wang et al., 2015*). Inhibition of ASIC1a or genetic knockout each result in a dramatic reduction in neuronal death following ischemic event (*Chassagnon et al., 2017*; *McCarthy et al., 2015*; *Pignataro et al., 2007*). It has long been suggested that ASIC1a may lead to cell death by contributing to $Ca^{2+}$ overload in neurons as this isoform has some permeability to $Ca^{2+}$ (*Xiong et al., 2004*). However, an intriguing newer hypothesis has suggested that ASIC1a can act as a receptor in addition to its function as an ion channel (*Wang et al., 2020*; *Wang et al., 2015*). This hypothesis posits that, during stroke, the pH of the extracellular solution falls, leading to a conformational change in the channel which exposes a binding site on the C-terminus for serine/threonine receptor interacting protein kinase 1 (RIPK1). RIPK1 is a cytosolic protein with multiple domains including an N-terminal kinase domain, an intermediate domain, and a C-terminal death domain (*Mifflin et al., 2020*). RIPK1 is known for its involvement in necroptosis or regulated necrotic cell death (*Galluzzi et al., 2017*). There are multiple ways to trigger necroptosis but the ensuing signaling cascade is relatively conserved. Generally, RIPK1, RIPK3 and mixed-lineage kinase domain-like pseudokinase (MLKL) are necessary to form the necrosome which is the required functional unit for cell death (*Wang et al., 2014*).

Given the hypothesis that ASIC1a is a cell death receptor, it would be very attractive to design drugs that stabilize the complex between the N- and C-termini or destabilize the C-terminal interaction with RIPK1. With this idea in mind, our goal in this study is to test the hypothesis that the N-terminal domain (NTD) and C-terminal domain (CTD) of ASIC1a may form a dynamic complex that changes with pH. To do this, we have adapted a transition metal ion FRET (tmFRET) approach which is ideally suited to measure distances on this length scale. First, we incorporate the non-canonical amino acid fluorophore, 3-((6-acetylnaphthalen-2-yl)amino)–2-aminopropanoic acid (L-ANAP) into various sites within the NTD and CTD to serve as the FRET donor (*Chatterjee et al., 2013*). We then measure the position of the L-ANAP relative to a metal ion that is either attached to the channel using a cysteine-reactive metal chelator or incorporated into the membrane via an NTA conjugated lipid (*Gordon et al., 2018*; *Zagotta et al., 2016*). Using this approach, we do not see evidence that the NTD and CTD are in close proximity during resting pH that would be consistent with an interaction between the two domains. In addition, we measure modest movements towards one another during acidification. To confirm this result with a direct measure of binding, we also used microscale thermophoresis (MST) to test for binding between peptides of the termini and see no binding in this assay as well. Finally, we measure FRET between L-ANAP in our cytosolic domains and metal incorporated into the plasma membrane to determine the rearrangements that these domains undergo during acidification. Based on our results, we speculate that proximity of the CTD to the membrane during resting conditions, rather than a stable complex between the NTD and CTD, may prevent RIPK1 binding.

## Results

To determine the conformational changes that the intracellular domains of ASIC1a undergo, we first needed to develop an approach well-suited to make this measurement in full-length channels in intact membranes. To do this, we employ transition metal ion FRET (tmFRET). tmFRET utilizes a transition metal ion, such as $Co^{2+}$ and $Cu^{2+}$, as a non-fluorescent FRET acceptor and a small fluorophore attached to the channel at a specific site as the donor (*Latt et al., 1972*; *Latt et al., 1970*; *Taraska et al., 2009*). The transition metal ion quenches the donor fluorophore in a highly distant dependent manner.

Our approach needed to satisfy a number of criteria. (1) We needed to be able to site-specifically attach a fluorophore that is appropriate for tmFRET to our channel. (2) We needed to have a method for incorporating metal ions into our channel. (3) The probes we used cannot be sensitive to changes in pH. (4) The distance range over which our probes work needed to correspond with the distances between the intracellular domains. To accomplish all of this we adapted multiple methods to suit our needs for developing short distance probes for ASICs.

## Site-specific labeling of ASICs with an unnatural amino acid

To site specifically incorporate a small molecule fluorophore compatible with FRET utilizing a transition metal, we opted to incorporate the fluorescent unnatural amino acid (UAA) 3-[(6-acetyl-2-naphthalenyl)amino]-L-alanine (L-ANAP) at sites in the channel. L-ANAP, a derivative of prodan, has several advantages for measuring short intramolecular distances within an ion channel. First, L-ANAP has a small side chain, similar in size to a tryptophan (*Figure 1A*). This reduces artifacts in FRET measurements associated with the large size of fluorescent proteins (FP) and organic dyes, as well as long linkers that attach these chromophores to the protein of interest. In addition, its modest spectral overlap with the absorbance spectra of transition metals makes it well suited as a donor for measuring FRET over shorter distances. L-ANAP has been used to measure conformational dynamics in multiple ion channels and other proteins (*Gordon et al., 2018*; *Kalstrup and Blunck, 2013*; *Shandell et al., 2019*; *Suárez-Delgado et al., 2023*; *Zagotta et al., 2016*).

We used the amber suppression method to attach L-ANAP at specific sites on the NTD and CTD of ASIC1a (*Figure 1A*; *Chatterjee et al., 2013*; *Lee et al., 2009*). This approach utilizes two plasmids. The first encodes the channel with a TAG stop codon engineered at the desired site of labelling. The second plasmid, developed by Peter Schultz's lab and termed pANAP, encodes 4 copies of a tRNA as well as an evolved tRNA synthetase that together allows for the loading of a tRNA complementary to the TAG codon with L-ANAP (*Chatterjee et al., 2013*). These two plasmids were co-transfected into CHO-K1 cells that were incubated in cell-permeable methyl ester version of L-ANAP (see Methods for details of the incorporation protocol).

For these studies we used ASIC1a from rat with a genetically encoded mCitrine attached to the CTD of the channel via a 13 amino acid linker (*Figure 1B*). Since many of our desired labelling sites were in the short NTD of the channel, we needed to ensure that we minimized any 'leak' expression where the channel expresses even in the absence of L-ANAP. Work on the Shaker potassium channel has shown that TAG codons in the N-terminus can lead to translation re-initiation at non-canonical start codons downstream of the TAG site (*Kalstrup and Blunck, 2015*). To minimize this potential problem, we made a number of conservative mutations in the first 90 base pairs (30 amino acids) of the channel to minimize the chance of alternate initiation (*Figure 1—figure supplement 1*). In addition, the CTD has four native cysteine residues present which were mutated to serine, allowing us to then reintroduce a single cysteine residue at a site of our choosing. All cysteine residues in the extracellular region are engaged in disulfide bonds and are also too far away to contribute to a FRET signal given the working distance of tmFRET.

*Figure 1C* shows representative images of rASIC1a-S24TAG with a single intracellular cysteine at C469. Robust mCitrine expression was only observed when both plasmids are transfected and L-ANAP is added to the media, indicating full-length channel expression. Omission of either L-ANAP from the media or pANAP from the transfection results in failure to produce full-length channels and an absence of any measurable mCitrine signal.

To further ensure that full-length channels are only produced in the presence of the UAA, western blots were performed containing a number of potential TAG positions in the NTD, re-entrant loop, and CTD of the channel. Positions were excluded if no channels were seen on the blot or if a substantial signal was detected in the absence of L-ANAP in the media. *Figure 2A* shows two western blots that contain four different TAG mutation sites along with our pseudo wild-type construct, C469 WT, which has a single intracellular cysteine at position 469 and no introduced TAG mutation. Position S485TAG shows no expression even in the presence of L-ANAP in the growth media. Position H515TAG shows channel expression in both the presence and absence of L-ANAP. These are each an example of positions we did not pursue further. S464TAG and M505TAG, however, show robust ASIC1a expression only when L-ANAP is present in the media making these suitable positions to move forward with.

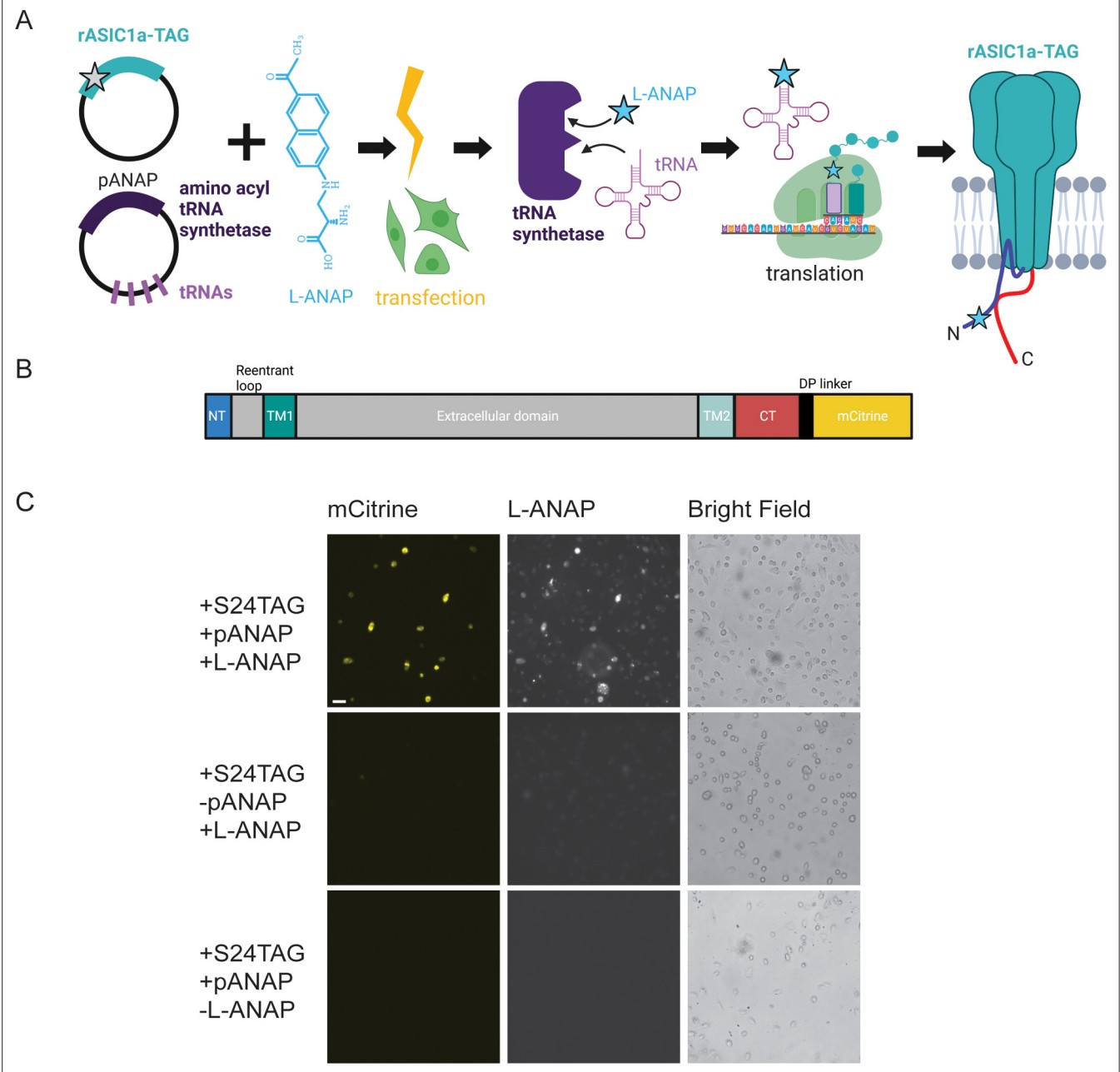

**Figure 1.** Incorporation of L-ANAP into ASIC1a. (**A**) Cartoon schematic illustrating L-ANAP incorporation into rASIC1a using the amber suppression method. Gray star indicates the location of the TAG mutation where L-ANAP can be incorporated when rASIC1a is co-transfected with pANAP and the media is supplemented with free L-ANAP (Created with Biorender.com). (**B**) Schematic of rASIC1a construct used for tmFRET experiments. Cytosolic N- and C-termini are in blue and red, respectively. mCitrine is conjugated to the C-terminus using a linker based on a loop region in DNA polymerase (DP). (**C**) Fluorescent and bright field images of rASIC1a-S24TAG expressed in CHO-K1 cells. Left column shows mCitrine fluorescence while the middle columns shows the L-ANAP fluorescence signal. Images were collected either in the presence or absence of L-ANAP and pANAP. Images were collected 24 hr after transfection at 20 x. The scale bar in the top left is 50 μm and applies to all panels in C.

The online version of this article includes the following figure supplement(s) for figure 1:

**Figure supplement 1.** DNA sequence for rASIC1a in black with conservative alternative initiation site mutations highlighted in cyan and cysteine to serine mutations highlighted in yellow.

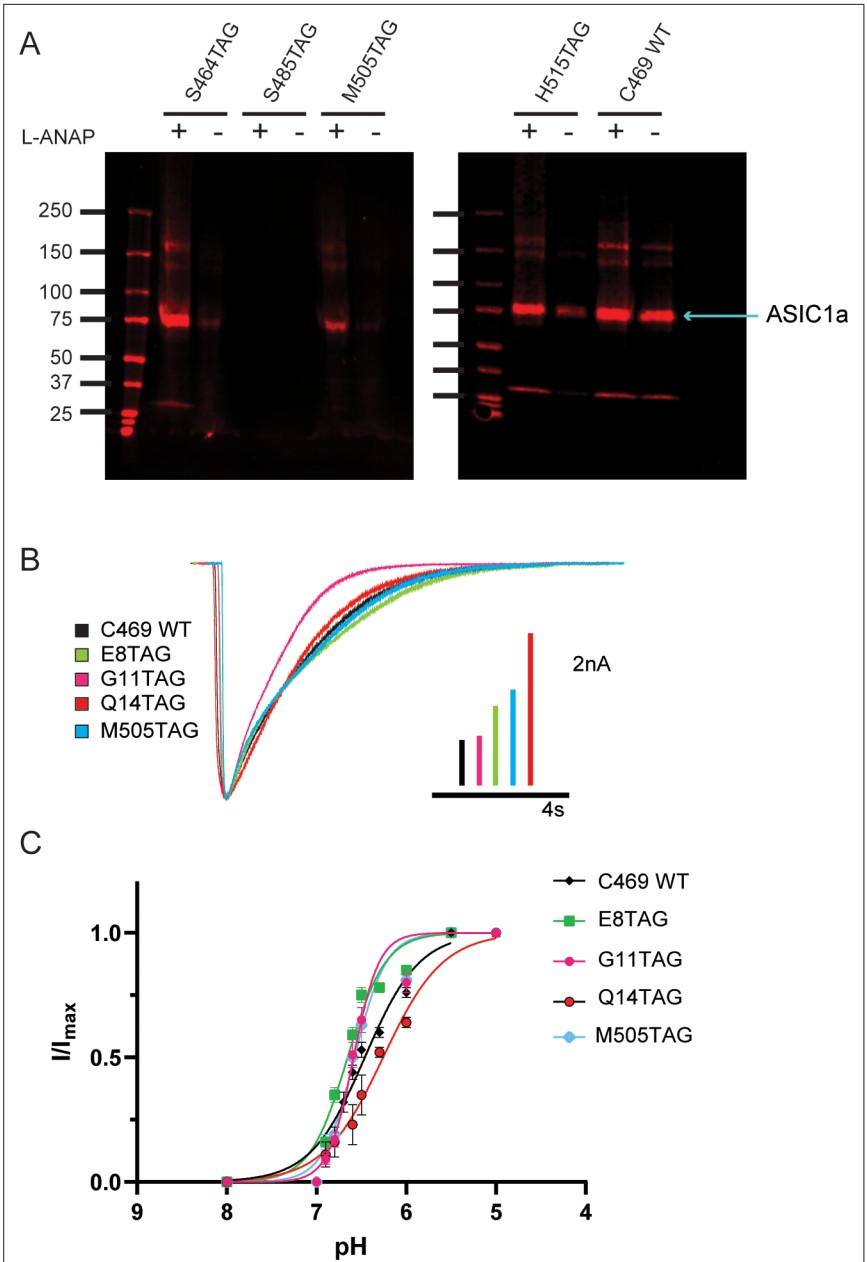

**Figure 2.** Incorporation of L-ANAP into ASIC1 produces full-length channel with normal pH-dependent gating.
(**A**) Representative western blots for rASIC1a expressed in CHO-K1 cells with a single TAG mutation at denoted
position, blotted with an anti-GFP antibody. Each TAG mutant was cultured both with L-ANAP (10 μM) and without
L-ANAP supplementation. Expression of channels with L-ANAP incorporated is seen with bands at ~75 kDa.
Higher molecular weight bands are likely channel oligomers (dimer, trimer) and bands below are likely free
mCitrine. Neither the higher nor lower bands make up a significant portion of channel expressed. Less protein
was loaded in the WT (no TAG) control C469 WT. (**B**) Representative whole-cell recordings of TAG mutants with
single cysteine at C469 (colors) and C469 WT construct (black) elicited by solution switch from pH 8 to pH 5.5.
(**C**) pH dependence of activation for rASIC1a TAG mutants with single cysteine at C469 (colors) and the C469 WT
construct (black). Data were fit to a modified Hill equation. Fit results found in *Figure 2—figure supplement 1*.

The online version of this article includes the following source data and figure supplement(s) for figure 2:

**Source data 1.** Source data for *Figure 2C* showing each cells individual fit $pH_{0.5}$ along with the mean and SEM and
P-values.

**Figure supplement 1.** Representative western blots for all TAG mutants created in this study.

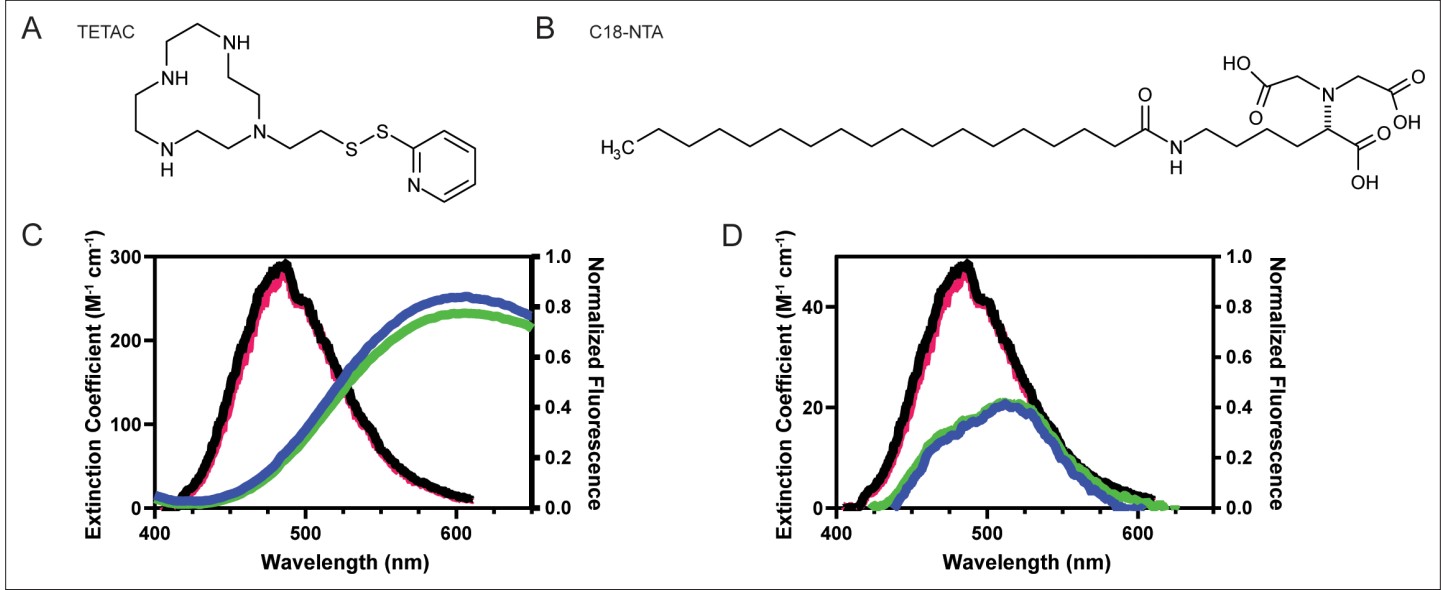

**Figure 3.** Spectral properties of L-ANAP and Cu$^{2+}$-TETAC and C18-NTA make both good FRET pairs capable of measuring short distances. (**A**) Chemical structure of free TETAC. (**B**) Chemical structure of C18-NTA. (**C**) Spectral properties of L-ANAP bound to agarose beads and Cu$^{2+}$-TETAC. Absorption spectra of TETAC-Cu$^{2+}$ at pH 8 (blue) and pH 6 (green) were measured and spectral properties plotted along with emission spectra from L-ANAP bound to agarose beads at pH 8 (black) and pH 6 (red). (**D**) Spectral properties of L-ANAP bound to agarose beads and Co$^{2+}$-C18-NTA. Absorption spectra of Co$^{2+}$-C18-NTA at pH 8 (blue) and pH 6 (green) were measured and spectral properties plotted along with emission spectra from L-ANAP bound to agarose beads at pH 8 (black) and pH 6 (red).

The online version of this article includes the following figure supplement(s) for figure 3:

**Figure supplement 1.** Spectral properties of Zn$^{2+}$-TETAC and Zn$^{2+}$-C18-NTA at pH 8 and pH 6.

Representative western blots for every position we tested can be seen in *Figure 2—figure supplement 1*.

## Incorporation of L-ANAP into ASIC1a does not dramatically alter channel function

We performed whole-cell patch clamp experiments for each position to ensure that the channel remained functional despite the incorporation of L-ANAP. Each mutant channel showed a robust current when L-ANAP was present in the media (*Figure 2B*). Each mutant showed the typical rapid activation upon a fast switch from pH 8 to pH 5.5 followed by channel desensitization. While rASIC1a-G11TAG showed a slightly altered desensitization rate, all mutants show complete desensitization in less than 10 s. In addition, we measured the pH dependence of channel activation. Our pseudo wild-type construct, C469 WT, showed a half-activating pH of 6.52 similar to our previously measured pH$_{0.5}$ for WT ASIC1a (*Klipp and Bankston, 2022*). The pH$_{0.5}$ values for the mutants all fell between 6.42 and 6.67 indicating that normal pH-dependent gating remains largely intact in each mutant (*Figure 2C*). The normal function of these mutant channels, we believe, makes these well-suited to continue examining rearrangements that occur in the NTD and CTD of the channel.

## pH does not alter L-ANAP fluorescence or metal ion absorbance

To measure conformational changes in these regions, we employed two different tmFRET methodologies. The first, termed ACCuRET (ANAP Cyclen-Cu$^{2+}$ resonance energy transfer), incorporates metal ions at specific sites on the channel using a cysteine reactive metal chelator, 1-(2-pyridin-2-yldisulfanyl) ethyl)–1,4,7,10-tetraazacyclododecane (TETAC, *Figure 3A*; *Gordon et al., 2018*). TETAC has a short linker connected to a cyclen ring that binds transition metals with a sub-nanomolar affinity (*Mutsuo Kodama et al., 1977*). The second method involved attaching the metal to the plasma membrane using a synthesized lipid with a metal-chelating head group called stearoyl-nitrilotriacetic acid (C18-NTA, *Figure 3B*; *Gordon et al., 2016*). We termed this method memFRET for clarity. Using TETAC and

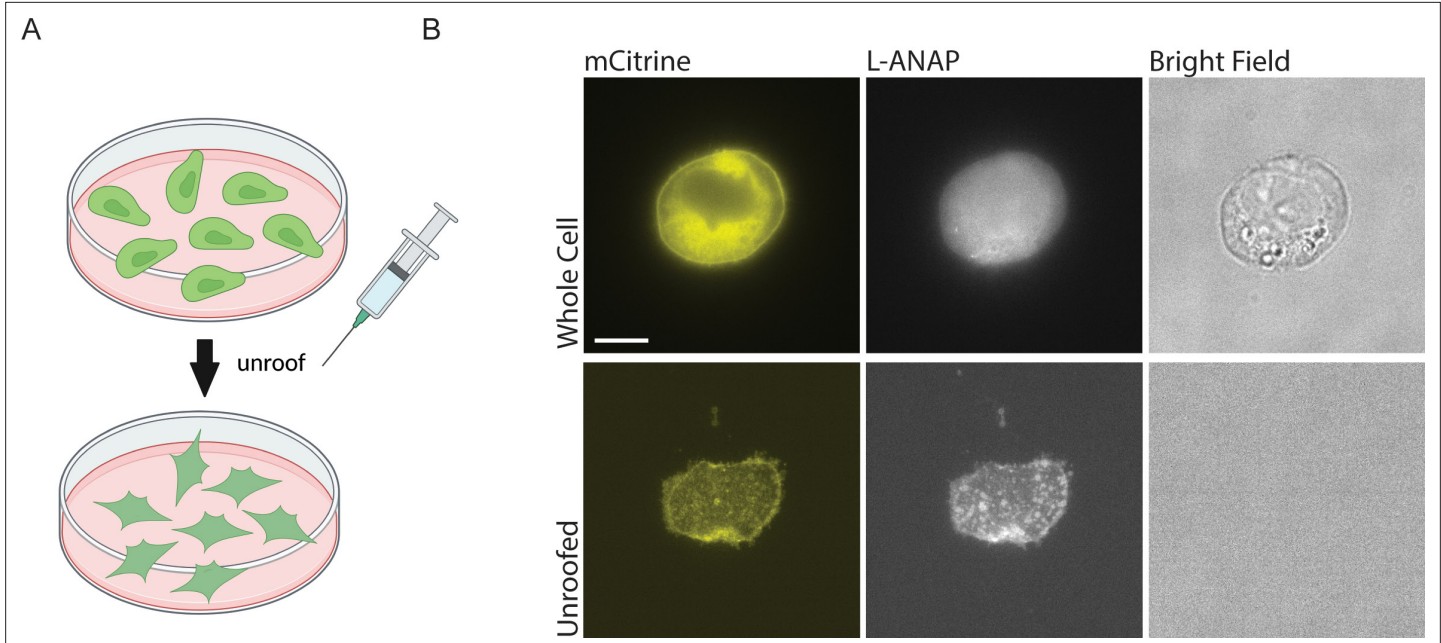

**Figure 4.** Cell unroofing isolates the plasma membrane for imaging. (**A**) Cartoon schematic of unroofing method used. Cultured CHO-K1 cells are washed with DPBS before being subjected to a constant stream of buffer using a needle and syringe to remove the non-adhered portion of the cells (Created with Biorender.com). (**B**) Representative fluorescent and bright field images at 60 x of whole cell (top) and unroofed (bottom) expressing rASIC1a-G11TAG. The scale bar in the top left is 10 µm and applies to all panels in B.

C18-NTA allowed us to introduce transition metal ions at a specific site within the protein (TETAC) or scattered across the plasma membrane (C18-NTA).

For each of these approaches we calculated the expected FRET efficiency between L-ANAP and the coordinated metal ion. *Figure 3C* shows the emission spectrum of L-ANAP along with the extinction coefficient of $Cu^{2+}$-TETAC. The extinction coefficient was determined by measuring the absorbance of $Cu^{2+}$-TETAC across the appropriate wavelengths (see methods for calculation). Similarly, *Figure 3D* shows the same data for $Co^{2+}$-NTA. Using these data and assuming a quantum yield of 0.22 (*Gordon et al., 2018*; *Gordon et al., 2016*) and $\kappa^2$ of 2/3 (*Fung and Stryer, 1978*; *Loura, 2012*), we calculate an $R_0$ for ACCuRET of 17.6 Å and for memFRET of 12.4 Å at pH 8 where we expect all the channels to be in the resting state (*Table 1*).

L-ANAP has been previously shown to not be pH sensitive when incorporated into the protein backbone (*Suárez-Delgado et al., 2023*). We confirmed that result here by mimicking incorporation of L-ANAP into the protein by conjugating L-ANAP to amine reactive agarose beads and measuring the emission spectra for L-ANAP on the same microscope setup used for the subsequent experiments. The emission spectrum of L-ANAP did not differ in either our rest (pH 8) or acidification (pH 6) buffers (*Figure 3C and D*). In addition, the extinction coefficient of $Cu^{2+}$-TETAC and $Co^{2+}$-NTA also do not change when the pH of the solution is changed (*Figure 3C and D* respectively). This results in a minimal change in $R_0$ values between pH 8 and pH 6 for ACCuRET (17.6 Å and 17.2 Å, respectively) and for memFRET (12.4 Å and 12.6 Å, respectively).

Finally, in order to apply either of these metal labels to the CTD or the inner leaflet of the plasma membrane, we needed access to the intracellular side of the channel. To do this, we used cell unroofing which was originally used for EM (*Heuser, 2000*) but has been adopted for tmFRET experiments as well (*Gordon et al., 2018*; *Gordon et al., 2016*; *Zagotta et al., 2016*). In short, we used a fluid stream to create a mechanical shearing force to remove the dorsal surface and all the intracellular components leaving only the ventral surface of the plasma membrane that is attached to the cover

**Table 1.** $R_0$ values calculated for $Cu^{2+}$-TETAC and $Co^{2+}$-C18-NTA using *Equation 3* at pH 8 and pH 6.

|  | pH 8 | pH 6 |
|---|---|---|
| ACCuRET | 17.6 | 17.2 |
| memFRET | 12.4 | 12.6 |

glass (*Figure 4A*). Before unroofing the cells, the L-ANAP signal is diffuse throughout the cell, without obvious membrane localization as expected, and the mCitrine signal can be seen in the membrane and in other intracellular membranes. After unroofing the cells, only fluorescence associated with the 'footprint', or ventral surface, of the cell membrane is left behind. L-ANAP and mCitrine fluorescence colocalized well in the unroofed cells (*Figure 4B*). All together, these data show that we were able to combine multiple approaches to site-specifically label ASIC1a with the small molecule fluorophore, L-ANAP, as well as metal ions and none of the probes we use were impacted by changes in pH. With these combined approaches, we can measure conformational changes that occur in the intracellular domains of the channel.

## Measuring the distance between the N- and C- termini

With these methodologies in place, we can directly measure rearrangements within the intracellular domains of ASIC1a. The ACCuRET approach (*Figure 5A*) is particularly well-suited for measuring the distance between the NTD and the CTD for several reasons. (1) The $R_0$ for our FRET pairs is on the order of ~17–18 Å which is ideal for measuring distances between the intracellular domains as the distance between the bottom of TM1 and TM2 in the channel is on the order of 10–20 Å. (2) We can measure these distances in intact channels embedded in native membranes. (3) The fluorophore and metal ion are small and have relatively few bonds connecting them to the protein backbone making them good proxies for the position of the NTD and CTD relative to one another.

*Figure 5B* shows a representative ACCuRET experiment for FRET between L-ANAP at rASIC1a-Q14TAG and $Cu^{2+}$-TETAC at C469. This first image shows the L-ANAP fluorescence before addition of the metal ion. The addition of 10 µM $Cu^{2+}$-TETAC- resulted in a significant reduction in the L-ANAP signal which was reversed upon application of 1 mM DTT. The normalized FRET efficiency was then calculated (see methods for details) and a summary of all the data measured between L-ANAP at rASIC1a-Q14TAG and $Cu^{2+}$-TETAC at C469 is shown in *Figure 5C*. The mCitrine signal does not show the same quenching and recovery as the L-ANAP which is expected but does photobleach over time.

Recall that the hypothesis that we are testing suggests that the NTD and CTD of ASIC1a form a complex at rest and that this complex is critical for preventing the initiation of cell death by the channel (*Figure 5D*). The putative interaction site between the intracellular domains was speculated to involve four consecutive glutamate residues at positions 6–9 on the NTD and residues 468–476 on the CTD (*Figure 5D*, inset) (*Wang et al., 2020*; *Wang et al., 2015*). Specifically, Rosetta modelling showed a very close (4.7 Å) proximity between residues E7 and K468 (*Wang et al., 2020*). This hypothesis predicts that the positions along the short NTD should show high FRET efficiency with metal ions incorporated into the channel at C469 at rest and that the FRET between the intracellular domains should decrease with prolonged (~minutes) acidification.

ACCuRET between L-ANAP at E8 and $Cu^{2+}$-TETAC at C469, the two residues immediately adjacent to the modelled close contact between the NTD and CTD, showed a modest 20% FRET efficiency putting the NTD and CTD ~22 Å apart which is significantly farther than the modelling predicted (*Figure 6A*, *Table 2*). However, it is possible that the presence of the labels in the binding site may impact the two domains ability to interact. To circumvent this, we measured distances between the NTD and CTD with a large combination of L-ANAP positions as well as $Cu^{2+}$-TETAC positions.

ACCuRET for two more positions along the NTD of ASIC1a (G11TAG and Q14TAG) relative to the same C469 have similar results to the E8TAG position. Modest FRET signals are measured again consistent with distances 20–22 Å range (*Figure 6A*, *Table 2*). Upon acidification, a slight increase in FRET efficiency is seen for all three positions, consistent with the NTD and CTD moving slightly closer together under these conditions. In addition, we moved the site of TETAC incorporation out of the putative binding site to residue 477 and remeasured ACCuRET between G11TAG and Q14TAG at this new position. Again, we measured only modest FRET efficiencies consistent with distances in the ~20 Å range and a trend towards increased FRET at pH 6 (*Figure 6B*, *Table 2*).

To confirm that the quenching we see is from the transition metals is due to FRET between L-ANAP and the $Cu^{2+}$-TETAC, we repeated some of these measurements loading TETAC with $Zn^{2+}$ instead of $Cu^{2+}$ (*Lacerda et al., 2007*). $Zn^{2+}$ does not absorb light in the visible spectrum and thus does not quench L-ANAP (*Figure 3—figure supplement 1*). As expected, no FRET is measured between L-ANAP and E8TAG or G11TAG and $Zn^{2+}$-TETAC at C469 (*Figure 6A*) indicating that the quenching we see in our experiments is due to the presence of the transition metal. Additionally, the lack of

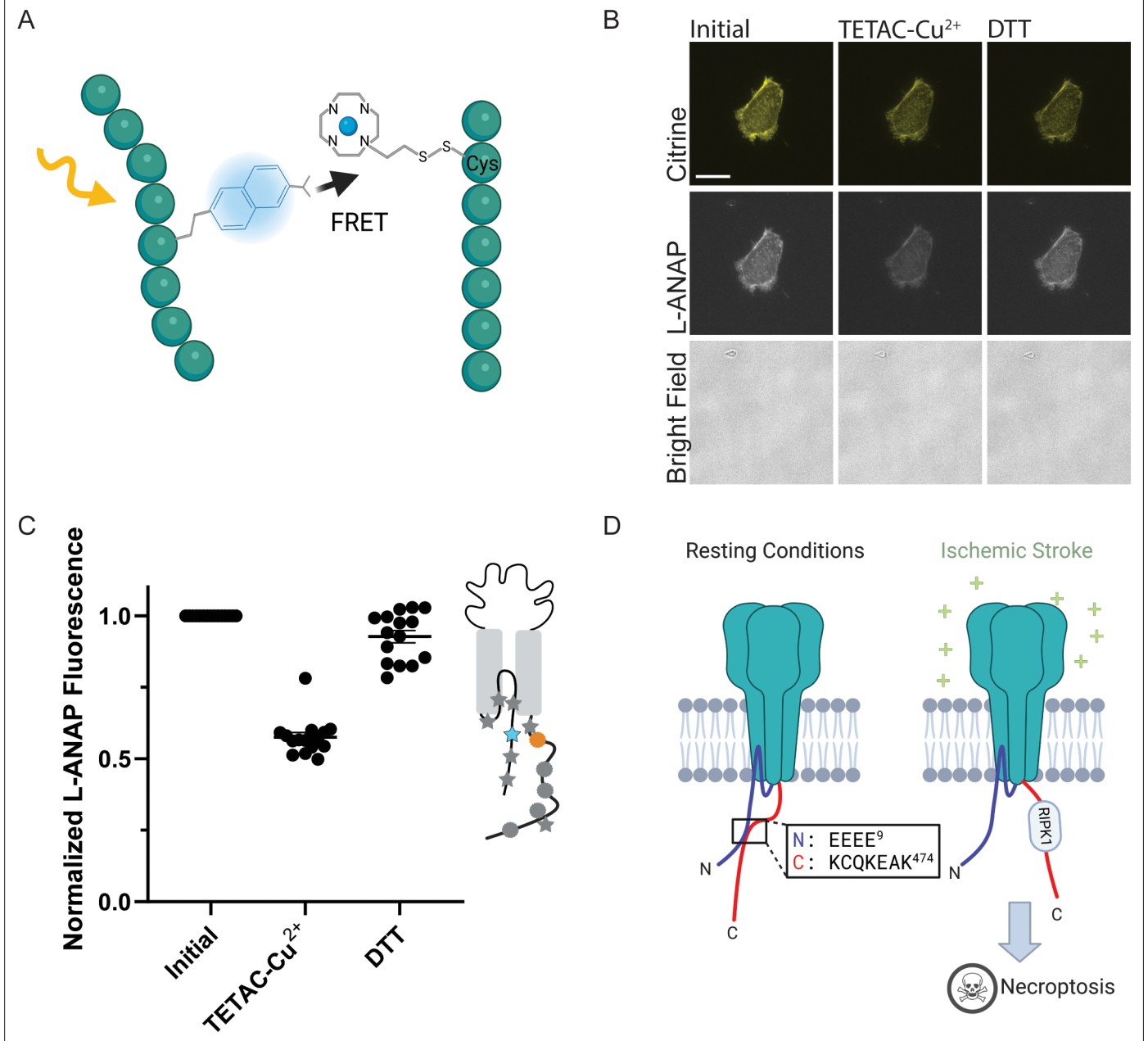

**Figure 5.** ACCuRET measures short-range distances between the NTD and CTD of ASIC1a. (**A**) Cartoon showing the ACCuRET approach (Created with Biorender.com). (**B**) Representative ACCuRET experiment. Fluorescence and bright field images at 60 x of rASIC1a-Q14TAG at pH 6. In brief, the first image is taken in control buffer at pH 6. This is followed by washing on TETAC preloaded with $Cu^{2+}$ and then washout of the excess $Cu^{2+}$-TETAC. The second image is then taken which shows a decrease in L-ANAP fluorescence indicative of FRET. Finally, $Cu^{2+}$-TETAC is removed by adding DTT and the final image is taken where the L-ANAP fluorescence is restored. mCitrine bleaches across all three images and the bright field is blank, indicating the cell is successfully unroofed. The scale bar in the top left is 10 μm and applies to all panels in B. (**C**) Normalized FRET efficiency for example experiment in B. Signals are normalized to fluorescence before addition of metal ions. The inset cartoon shows the position of L-ANAP denoted by a blue star and the single cysteine denoted by an orange circle. Grey symbols represent other positions that are mutated in this study in order to show the relative position mutated in this example. In this instance, L-ANAP is incorporated at Q14 and the single cysteine in the C-terminus is at position C469. (**D**) Cartoon illustrating the current hypothesis of the orientation between the NTD and CTD at rest (left) and after prolonged acidosis (right). The inset sequence shows the putative interaction between the intracellular domains.

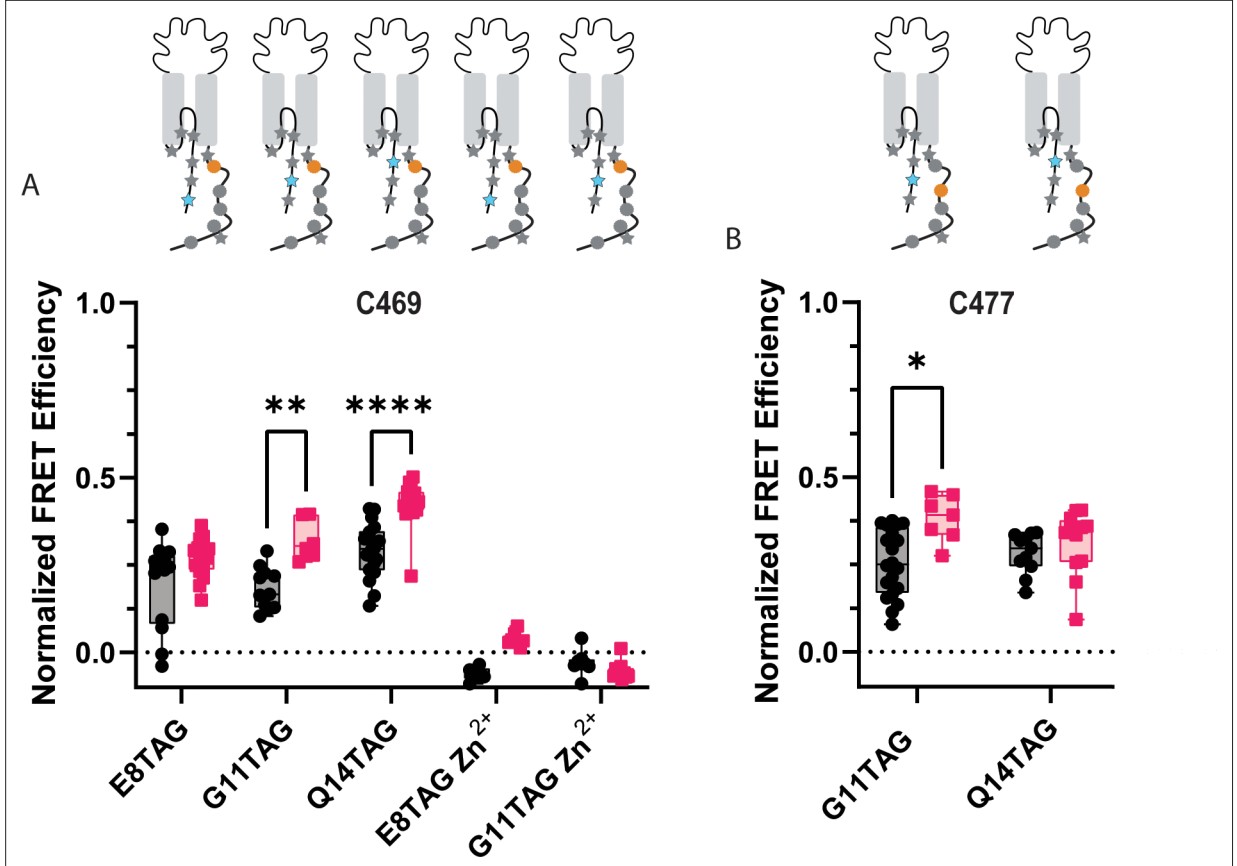

**Figure 6.** ACCuRET reveals the NTD and CTD likely do not form a complex at rest. (**A**) Normalized FRET efficiency between L-ANAP on the NTD and Cu$^{2+}$-TETAC at position C469 at pH 8 (black) and pH 6 (red). FRET efficiencies, SEM, N and calculated distances are summarized in **Table 2**. Zn$^{2+}$-TETAC negative control for positions E8 and G11 shows that the changes in signal measured with Cu$^{2+}$-TETAC are due the presence of Cu$^{2+}$ and not due to any other changes. (**B**) Normalized FRET efficiency between L-ANAP on the NTD and Cu$^{2+}$-TETAC at position C477 at pH 8 (black) and pH 6 (red). The inset cartoons show the position of L-ANAP denoted by a blue star and the single cysteine denoted by an orange circle. Box and whisker plot with whiskers ranging from minimum to maximum values and the bottom and top edges of the box denoting the 25th and 75th quartiles, respectively. Statistical significance shown using two-way ANOVA with Tukey's multiple comparisons test are denoted between relevant comparison using appropriate asterisks. Ns indicates $p > 0.05$, * indicates $p \leq 0.05$, ** indicates $p \leq 0.01$, *** indicates $p \leq 0.001$, **** indicates $p \leq 0.0001$. $6 < N < 21$ for conditions shown here. **Table 2** and source data show exact N for each condition.

The online version of this article includes the following source data for figure 6:

**Source data 1.** P-Values of relevant comparisons between NTD TAG positions with cysteine at C469 and C477 at pH 8 and pH6.

measured FRET in our Zn$^{2+}$ control precludes the possibility that FRET between L-ANAP and mCitrine convolute our tmFRET measurements.

It is possible that the CTD binding site for the NTD is at a more distal position along the CTD than originally speculated. To test this, we again moved the Cu$^{2+}$-TETAC-labelled cysteine to residues further along the CTD. Cu$^{2+}$-TETAC at C485 shows similar FRET at pH 8, but, interestingly, shows a larger movement (~6 Å) towards the NTD at pH 6 (**Figure 7A**). Again, this movement towards one another is likely inconsistent with a model where the termini are in complex at rest and move apart during acidification. Finally, the most distal positions we measured (C505, C515), also display modest FRET signals at pH 8 with 515 showing a larger (~5 Å) movement towards the NTD upon acidification (**Figure 7B and C**, **Table 2**).

In addition to the three positions on the cytosolic portion of the NTD, we also incorporated L-ANAP into three positions into the re-entrant loop of the channel. At the onset of this project, these positions were thought to be part of the cytosolic region as well, but newer Cryo-EM structures have demonstrated that residues ~19–41 are part of a loop that reinserts itself into the plasma membrane (**Yoder and Gouaux, 2020**). In each case, the FRET efficiency at rest is slightly higher than the efficiencies we

**Table 2.** Mean normalized FRET efficiency ± SEM for each TAG position relative to each cysteine (ACCuRET) or the membrane (memFRET) measured at pH 8 and pH 6.
Distances were calculated with *Equation 4*.

| | TETAC Position | TAG position | pH 8 | | | | pH 6 | | | |
|---|---|---|---|---|---|---|---|---|---|---|
| | | | mean normalized FRET efficiency | SEM | N | Distance (Å) | mean normalized FRET efficiency | SEM | N | Distance (Å) |
| ACCuRET | | E8TAG | 0.20 | 0.03 | 13 | 22 | 0.27 | 0.01 | 17 | 20 |
| | | G11TAG | 0.18 | 0.02 | 11 | 23 | 0.32 | 0.02 | 6 | 20 |
| | | Q14TAG | 0.29 | 0.02 | 18 | 20 | 0.43 | 0.02 | 15 | 18 |
| | C469 | S24TAG | 0.45 | 0.02 | 10 | 18 | 0.57 | 0.01 | 8 | 16 |
| | | I33TAG | 0.37 | 0.02 | 21 | 19 | 0.58 | 0.01 | 17 | 16 |
| | | A44TAG | 0.43 | 0.02 | 6 | 19 | 0.27 | 0.02 | 14 | 20 |
| | | G11TAG | 0.26 | 0.02 | 20 | 21 | 0.38 | 0.03 | 7 | 19 |
| | C477 | Q14TAG | 0.28 | 0.02 | 11 | 21 | 0.31 | 0.03 | 12 | 20 |
| | | S24TAG | 0.35 | 0.02 | 22 | 20 | 0.49 | 0.04 | 9 | 17 |
| | | E8TAG | 0.11 | 0.02 | 22 | 25 | 0.26 | 0.03 | 13 | 21 |
| | C485 | G11TAG | 0.28 | 0.02 | 14 | 21 | 0.59 | 0.02 | 10 | 16 |
| | | Q14TAG | 0.21 | 0.04 | 27 | 22 | 0.58 | 0.02 | 23 | 16 |
| | | S24TAG | 0.32 | 0.04 | 16 | 19 | 0.57 | 0.01 | 13 | 16 |
| | C505 | E8TAG | 0.27 | 0.01 | 12 | 21 | 0.37 | 0.02 | 13 | 19 |
| | | Q14TAG | 0.41 | 0.02 | 6 | 19 | 0.43 | 0.03 | 13 | 18 |
| | C515 | E8TAG | 0.08 | 0.01 | 14 | 26 | 0.24 | 0.01 | 9 | 21 |
| | | Q14TAG | 0.20 | 0.02 | 7 | 22 | 0.45 | 0.01 | 7 | 18 |
| ACCuRET Zn$^{2+}$ controls | C469 | E8TAG | –0.06 | 0.01 | 6 | N/A | 0.04 | 0.01 | 8 | N/A |
| | | G11TAG | –0.03 | 0.01 | 8 | N/A | –0.05 | 0.01 | 10 | N/A |
| memFRET | N/A | E8TAG | 0.42 | 0.03 | 11 | 10 | 0.40 | 0.03 | 14 | 11 |
| | | G11TAG | 0.46 | 0.03 | 9 | 10 | 0.48 | 0.01 | 15 | 9 |
| | | Q14TAG | 0.44 | 0.02 | 10 | 10 | 0.60 | 0.03 | 7 | 6 |
| | | S24TAG | 0.74 | 0.04 | 4 | ≤5 | 0.72 | 0.02 | 24 | ≤5 |
| | | I33TAG | 0.52 | 0.02 | 8 | 8 | 0.71 | 0.03 | 9 | ≤5 |
| | | S464TAG | 0.69 | 0.02 | 15 | ≤5 | 0.63 | 0.02 | 10 | ≤5 |
| | | M505TAG | 0.75 | 0.02 | 8 | ≤5 | 0.48 | 0.02 | 14 | 9 |
| MemFRET Zn$^{2+}$ controls | N/A | E8TAG | –0.13 | 0.01 | 8 | N/A | –0.15 | 0.02 | 9 | N/A |
| | | G11TAG | –0.03 | 0.01 | 8 | N/A | 0.01 | 0.01 | 9 | N/A |
| | | S464TAG | 0.06 | 0.04 | 6 | N/A | 0.01 | 0.03 | 10 | N/A |
| | | M505TAG | –0.12 | 0.04 | 6 | N/A | –0.12 | 0.03 | 9 | N/A |

measured between the NTD and CTD suggesting perhaps that the CTD might be somewhat closer to the re-entrant loop, and the membrane, than it is to the NTD (*Figure 7—figure supplement 1*). Measurements at pH 6 reveal that the CTD moves closer to the re-entrant loop with the notable exception of the position at A44TAG which is the only position we measure that shows a modest decrease in FRET with acidification.

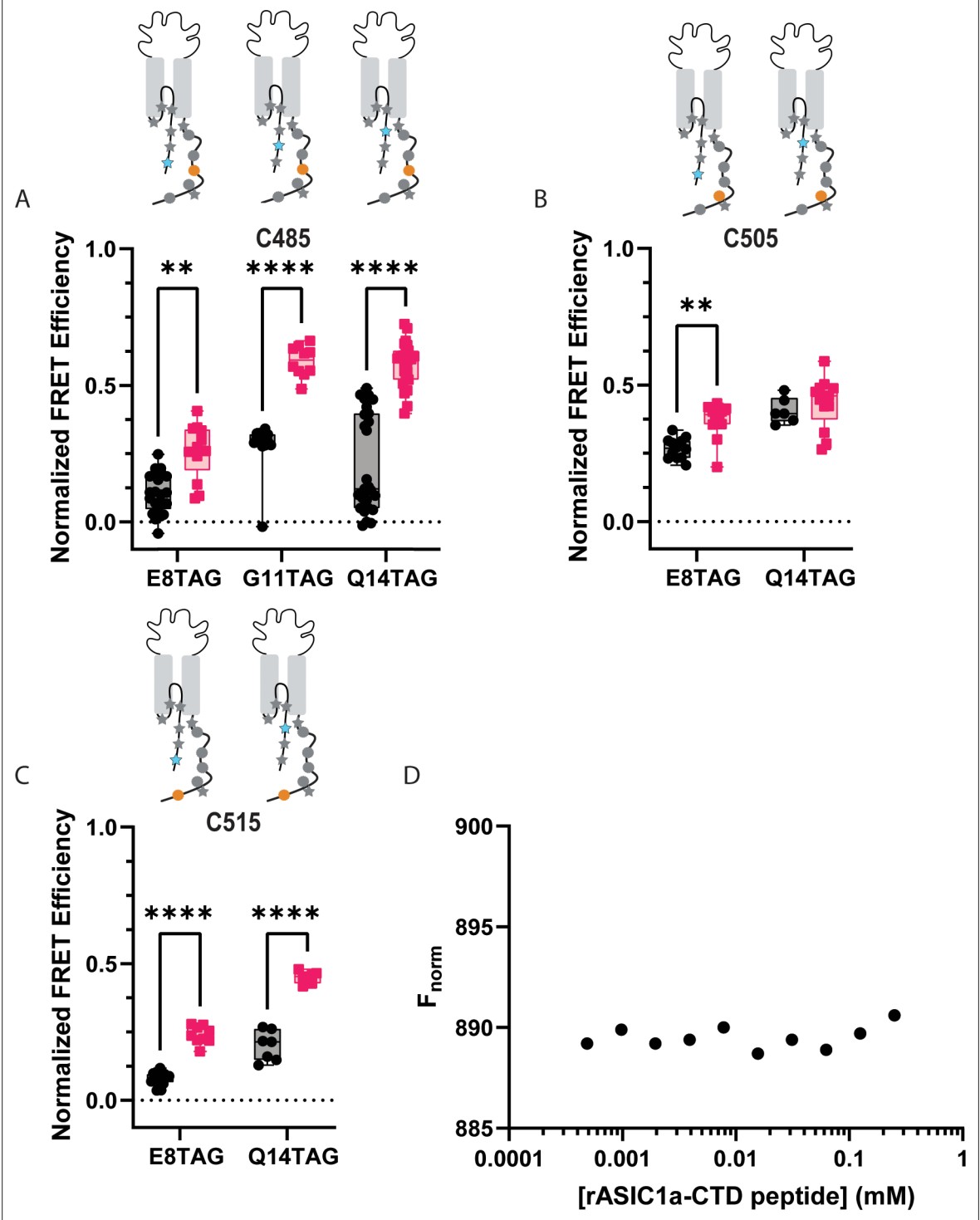

**Figure 7.** ACCuRET does not suggest an interaction between the NTD and more distal positions on the CTD. (**A**) Normalized FRET efficiency between L-ANAP on the NTD and $Cu^{2+}$-TETAC at position C485 at pH 8 (black) and pH 6 (red). FRET efficiencies, SEM, N and calculated distances are summarized in *Table 2*. (**B**) Normalized FRET efficiency between L-ANAP on the NTD and $Cu^{2+}$-TETAC at position C505 at pH 8 (black) and pH 6 (red). (**C**) Normalized FRET efficiency between L-ANAP on the NTD and $Cu^{2+}$-TETAC at position C515 at pH 8 (black) and pH 6 (red). (**D**) Representative binding curve for peptides of the NTD and CTD generated using MST. No indication of binding up to 250 µM. Box and whisker plot with whiskers ranging from minimum to maximum values and the bottom and top edges of the box denoting the 25th and 75th quartiles, respectively. Statistical significance shown using two-way ANOVA with Tukey's multiple comparisons test are denoted between relevant comparison using appropriate asterisks.

*Figure 7 continued on next page*

*Figure 7 continued*

Ns indicates p>0.05, * indicates p≤0.05, ** indicates p≤0.01, *** indicates p≤0.001, **** indicates p≤0.0001. 6<N<27 for conditions shown here. **Table 2** and source data show exact N for each condition.

The online version of this article includes the following source data and figure supplement(s) for figure 7:

**Source data 1.** P-Values of relevant comparisons between NTD TAG positions with cysteine at C485, C505, and C515 at pH 8 and pH6.

**Figure supplement 1.** FRET between the reentrant loop and the CTD.

## Peptides of the N- and C-termini do not bind in solution

While ACCuRET is a powerful tool for looking at changes in intracellular domains in intact channels in real membranes, it is not a direct measure of binding. To look for direct interaction between the NTD and CTD, we employed microscale thermophoresis (MST; *Figure 7D*). MST is an immobilization-free, solution-based method that measures biomolecular interactions (*Wienken et al., 2010*). This approach measures the motion of a fluorescent molecule along a microscopic temperature gradient created by an IR laser. This motion depends on a number of factors including the size and chemical environment of the molecule. Any binding between the molecule attached to the fluorophore and second molecule result in a change in the motion. To perform MST, we synthesized peptides for the NTD and CTD of ASIC1a from rat. The NTD peptide contained residues 1–17 with an additional C-terminal cysteine to which Alexa-647 was conjugated. The CTD peptide contained residues 460–526 with the endogenous cysteine residues mutated to serine. The labelled NTD peptide was kept at a constant 2.5 nM concentration with increasing amounts of the CTD. Little to no change in fluorescence signal was seen at any concentration of CTD. Using this approach, we see no signs of binding between the two domains up to 250 µM providing additional evidence that a complex between these two domains does not form.

## memFRET suggest termini reside in close proximity to the plasma membrane

Since our data suggest that the intracellular domains of the channel likely do not bind at rest and subsequently unbind during prolonged acidification, we decided to determine the topology of the NTD and CTD and then examine what sort of rearrangements these domains make during acidification. To do this, we employed a second tmFRET method that again uses L-ANAP as the FRET donor and metal ions as FRET acceptors but with the metal ions coordinated by membrane imbedded lipids instead of conjugated onto the channel which we termed memFRET (*Figure 8A*). By doing this, we measure distances between a single distinct L-ANAP position within a protein relative to the plasma membrane. We utilize a lipid with an NTA head group (stearoyl-nitrilotriacetic acid, C18-NTA) which introduces high-affinity metal binding sites throughout the plasma membrane. The $R_0$ for this pair is shorter than seen in ACCuRET (12.4 Å at pH 8) and distances are calculated taking into consideration the presence of multiple acceptors (see Methods). We found no difference in the distance between E8TAG, G11TAG, and Q14TAG and the plasma membrane under resting conditions suggesting that the NTD may lay parallel to the membrane. L-ANAP at Q14TAG moved nearly ~4 Å closer at pH 6 but we did not detect a change at the other two positions (*Figure 8B*, *Table 2*). Not surprisingly, L-ANAP at S24TAG and I33TAG, which are within the bilayer, both showed very large FRET efficiency with metal in the membrane with I33TAG displaying a lower FRET efficiency at rest compared to S24TAG at both pH 8 and pH 6, as well as itself at pH 6 (*Figure 8C*). A number of the FRET efficiency values that we measured here were so large that they exceeded the resolvable range of memFRET. While we cannot determine a precise distance from the membrane, they are likely in very close proximity. (*Table 2*).

Finally, to look at rearrangements of the CTD during acidification, we introduced TAG codons at two positions along the length of the CTD: 464 near the membrane and 505 nearer the end of the CTD. Surprisingly, at pH 8, both positions showed large FRET signals suggesting that the longer CTD, including the distal portion, may dwell near the plasma membrane (*Figure 8D*, *Table 2*). During acidification, however, the CTD appears to begin to move away from the plasma membrane with the more distal position moving farther than the more proximal position.

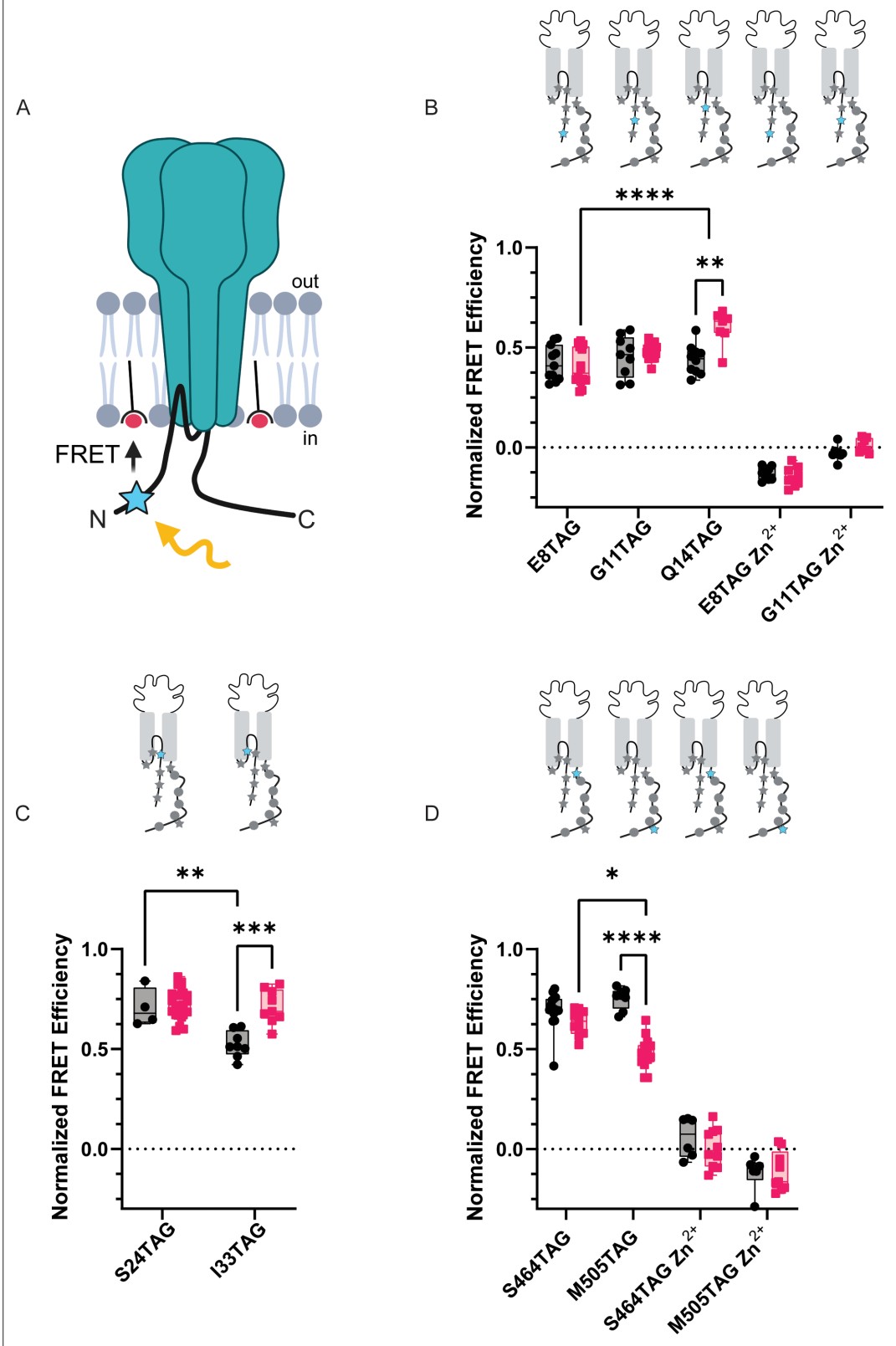

**Figure 8.** memFRET reveals that the NTD and CTD make frequent sojourns near the plasma membrane.
(**A**) Cartoon illustrating memFRET in unroofed cells between L-ANAP incorporated into the protein backbone and multiple C18-Co²⁺ incorporated into the plasma membrane (Created with Biorender.com). (**B**) Normalized FRET efficiency from memFRET experiments between L-ANAP at two sites in the re-entrant loop very near the plane

*Figure 8 continued on next page*

*Figure 8 continued*

of the intracellular leaflet of the plasma membrane. (**C**) Normalized FRET efficiency from memFRET experiments between L-ANAP on the NTD and $Co^{2+}$-C18-NTA at pH 8 (black) and pH 6 (red). $Zn^{2+}$-C18-NTA negative control for positions E8TAG and G11TAG shows that the changes in signal measured with $Co^{2+}$-C18 are primarily due the presence of $Co^{2+}$ and not to any other changes. (**D**) Normalized FRET efficiency from memFRET experiments between L-ANAP on the CTD and $Co^{2+}$-C18-NTA at pH 8 (black) and pH 6 (red). Each data point represents FRET efficiency measured within a single cell at either pH 8 in black or pH 6 in red. FRET efficiencies, SEM, N and calculated distances are summarized in *Table 2*. Statistical significance shown using two-way ANOVA with Tukey's multiple comparisons test are denoted between relevant comparison using appropriate asterisks. Ns indicates $p > 0.05$, * indicates $p \leq 0.05$, ** indicates $p \leq 0.01$, *** indicates $p \leq 0.001$, **** indicates $p \leq 0.0001$. $6 < N < 15$ for conditions shown here. *Table 2* and source data show exact N for each condition.

The online version of this article includes the following source data for figure 8:

**Source data 1.** p-Values of relevant comparisons between FRET involving the NTD or CTD TAG positions with metal ions in the plasma membrane at pH 8 and pH6.

## Discussion

The major aim of this work is to evaluate the hypothesis that the NTD of ASIC1a binds to the CTD in order to occlude a potential binding site for the cell death protein, RIPK1. Previous Rosetta modeling of the intracellular domains suggested that 4 consecutive glutamate residues in the NTD, E6-E9, electrostatically interact with positively charged residues in the CTD, K468, K471, K474 (*Wang et al., 2020*). The closest distance measured in the modeled closed state was 4.7 Å between E7 and K468. The modelled open state shows that the NTD and CTD are separated, suggesting that acidification leads to a conformational change in the intracellular domains that moves them apart. Moreover, experimental data using spectral FRET between CFP and YFP fused to the ends of each terminus observed a FRET efficiency of ~30% at rest and ~10% at pH 6 (*Wang et al., 2020*). The authors interpreted this FRET signal as evidence for complex formation and the decrease in FRET efficiency as an indicator that the termini moved apart during prolonged acidification.

Following up on these previous studies, we used the amber suppression method to incorporate L-ANAP throughout the NTD and CTD of ASIC1a. We then used these labelled channels to measure conformational changes using two tmFRET-based methods, ACCuRET and memFRET. To our knowledge, this is the first study to combine these two approaches to examine the structural rearrangement within a single protein. Our results show that the NTD and CTD of the channel sit ~18–26 Å apart on average and make small movements towards one another during acidification. The simplest interpretation of these results is that the intracellular domains do not form a stable complex at rest that

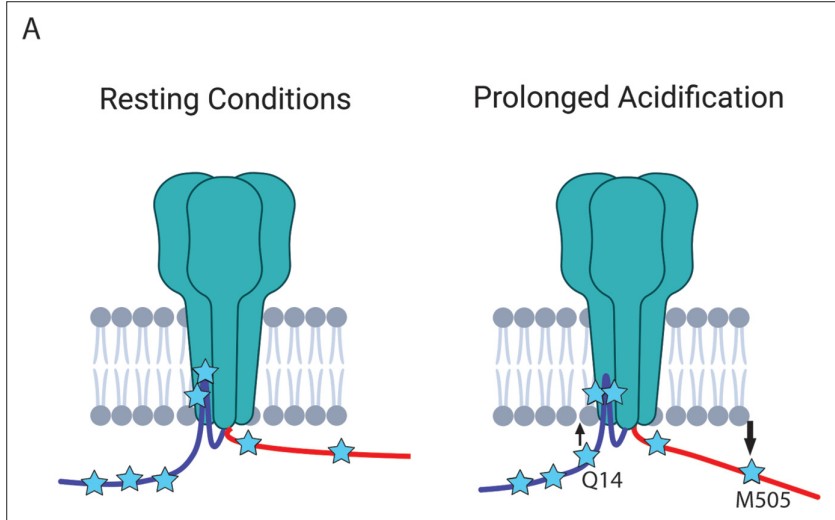

**Figure 9.** The NTD and CTD of ASIC1a likely do not interact. (**A**) Cartoon schematic of axial movements of the N- and C-termini relative to the membrane at rest and prolonged acidification. The proximal NTD moves slightly closer to the membrane while the proximal CTD moves away during acidification.

unbinds during acidification and that a new hypothesis for how acidification controls RIPK1 binding to ASIC1a is needed.

Given these results, we decided to look at dynamic rearrangements within the intracellular domains that might be occurring in order to potentially formulate alternative hypotheses for RIPK1 binding to ASIC1a. As expected, our memFRET measurements showed a large FRET efficiency between $Co^{2+}$ in the membrane and L-ANAP at S24TAG. This position is very near the bottom of the transmembrane segments in the plane of the inner leaflet providing a reasonable measure of the maximum FRET measurable in this assay. Looking, then, at our 3 positions in the cytosolic NTD we found comparable FRET efficiencies for each position consistent with the idea that the NTD might lay parallel to the plasma membrane, ~10 Å away from the inner leaflet, at rest. During acidification we detect a small move of the proximal NTD towards the plasma membrane.

Making the same measurements with L-ANAP at positions along the CTD, we found that position S464TAG, very near the bottom of transmembrane domain 2, shows an expected large FRET signal similar to S24TAG. Surprisingly, M505TAG showed large FRET with metal in the membrane, consistent with the idea that parts of the CTD may reside in close proximity to the plasma membrane as well. During exposure to pH 6 we see a decrease in FRET efficiency, especially at the more distal position, suggesting that during prolonged acidification the CTD may move away from the membrane (*Figure 9*).

## Comparison to previous studies

There have been two previous studies that have looked at potential rearrangements in the intracellular domains of ASIC1a. The first used CFP and YFP attached to the NTD and CTD and measured a change in the ratio of YFP/CFP intensity over time as a measure of FRET between the two domains. They found a slow pH dependent decrease in that ratio (*Wang et al., 2020*). The second study repeated this measurement with FPs that are not sensitive to pH in the pH 6–8 range and found that there was no change in the ratio of acceptor/donor fluorescence with prolonged acidification (*Couch et al., 2021*). YFP has a pKa of around 6.9 and its fluorescence changes dramatically between pH 7.4 and 6 (*Llopis et al., 1998*).

Additionally, the first study used spectral FRET between CFP and YFP fused to the ends of each terminus and observed a FRET efficiency of ~30% at rest and ~10% at pH 6. These efficiencies would suggest distances in the 55–70 Å range between the intracellular domains which is unlikely given the close proximity of the two transmembrane segments (10–20 Å based on published structures). Potential problems with these measurements likely stem from the inherent limitations to measuring intramolecular rearrangements over small distances with FP-based FRET. The classical FRET pairs have long $R_0$ values (30–60 Å), long flexible linkers (~15 Å), and are large in size (15–30 Å; *Best et al., 2007*; *Posson and Selvin, 2008*). Given this, it is hard to measure short distances and small rearrangement with these probes.

As stated above, the same group used Rosetta modelling to show that the termini may get as close together as 4.7 Å between E7 and K468. Our measurements suggest that the termini likely do not get closer than ~18 Å and may move closer during acidification, not farther apart. More specifically, we measured tmFRET between E8TAG and C469 which are the residues immediately adjacent to E7 and K468 and saw only a 20% FRET efficiency at pH 8 consistent with a distance of 22 Å apart.

The second study also used fluorescence lifetime imaging to show FRET between FP-tagged ASIC1a termini (*Couch et al., 2021*). These measurements showed that the FRET primarily originated from intra-subunit proximity and not from proximity between subunits. In addition, the structure shows that the intrasubunit distances between TMs are closer than the inter-subunit ones as well. While our data do not directly weigh in on the question of intra- vs intersubunit proximity, based on these two reasons, we would suggest that the FRET we measure is likely from intra-subunit FRET as well.

This same study mapped the topography of the intracellular termini of ASIC1a comparable to our memFRET experiments (*Couch et al., 2021*). In these experiments, they used the membrane soluble small molecule DPA as a FRET acceptor with CFP or YFP attached either to the end of the NTD or at 4 different positions along the length of the CTD (*Couch et al., 2021*). The results from the NTD are consistent with our findings here. They suggest that the most distal part of the NTD is ~6–10 Å from the membrane at rest and moves slightly closer at pH 6. This is consistent with our observation

that the most distal position we measure is ~10 Å away from the membrane and moves slightly closer during acidification.

Their results from the CTD quenching experiments are somewhat in contrast to what we see here. Our results suggest that the distal portion of the CTD spends significant time near the plasma membrane: enough to generate the large quenching signal we measure. The DPA quenching experiments suggest the proximal portion of the CTD is ~12–16 Å from the plasma membrane and the more distal portions extend down into the cytosol. Determining the motion of the CTD is complicated by the biphasic nature of the distance dependence curves for DPA quenching. The authors conclude that the CTD moves towards the membrane during acidification but acknowledge that their data could also be consistent with a move away, similar to what we see. There are a number of potential sources of the differences in our measurements. First, the previous study used ASIC1 from chicken whereas we used ASIC1a from rat. The sequences of the termini are largely conserved but the conformational changes that surround prolonged acidification could differ. It is well established, for instance, that ASIC1a from rat (and human) undergo a slow inactivation process during acidosis called tachyphylaxis that is absent in the chicken isoform (*Chen and Gründer, 2007*; *Rook et al., 2020*). Second, insertion of large FPs into a relatively short CTD could impact the ability of the CTD to position itself near the plasma membrane.

## Limitations to our approach

It is important to note that our approaches has limitations that need to be considered when thinking about our results. A number of controls are required when attempting to incorporate UAAs. The western blots and function give us confidence that we are primarily measuring full-length ANAP-incorporated ASIC1a. Given the position of the N-terminal TAG mutations, premature truncation at any of these positions would yield very short peptides that are not capable of forming functional channels. We only see bands in the presence of -LANAP in our western blots suggesting that read-through, where the ribosome inserts a natural amino acid instead of L-ANAP, should not present a major problem. However, it is possible that premature truncation at positions in the CTD yield truncated subunits capable of forming functional channels. These truncated subunits could co-assemble with other truncated subunits and full-length L-ANAP containing subunits. The truncated channels would be invisible to the memFRET measurement but it is possible that co-assembled truncated channels could alter the position of the CTD.

It is also possible that the bulky FP on the CTD could impact the ability of the termini to interact. We do not think this is likely the case. First, previous work looking at the interaction between the N- and C-termini used the same bulky FPs (*Couch et al., 2021*; *Wang et al., 2020*). In those experiments there was often an FP on each terminus. Second, our tag was attached following an 11-residue linker which should give enhanced flexibility and reduced steric hindrance. We previously showed that ASIC3 channels containing mCerulean after this linker were able to bind the protein Stomatin at a binding cite in the distal CTD near the FP (*Klipp et al., 2020*). Given that the NTD is significantly smaller than Stomatin, we believe there is likely sufficient room for the complex to still form.

ACCuRET has a number of potential complications, some of which apply to our system. First, unincorporated L-ANAP, or any background fluorescence, can contaminate our fluorescence signals and alter our measurements. In cells that do not express mCitrine, we see little to no L-ANAP signal, suggesting that this is not likely a big contributor to our measurements. Second, it is not possible to verify complete labelling of the channel with either L-ANAP or TETAC. Third, in order to perform these experiments, we needed to unroof the cells to have access to the CTD cysteine residues. In doing so, we expose both sides of the membrane to the changes in pH. It is possible that acidification of the intracellular side of the channel alters the normal positions and conformational changes of these domains. However, prolonged extracellular acidification can lead to drops in intracellular pH as well so our approach may not differ as much as expected from external application (*Doyen et al., 2022*; *Salameh et al., 2014*). Additionally, unroofing could disrupt the interaction between the NTD and CTD by washing away other proteins that are critical to maintain this interaction. Finally, FRET is highly non-linear with the closest approach distances most represented in the FRET signal. It is likely that the intracellular domains are largely unstructured and dynamic. Our distance measurements are more representative of an average position between the domains rather than a fixed distance and that average position is skewed towards the closest approach between the donor and acceptor. However,

this limitation, we believe, strengthens our hypothesis that the intracellular domains are likely not in close proximity to one another at rest. Given this averaging of the position of the termini, it is possible that the termini exist in close proximity for a very small proportion of time. However, if this interaction is crucial for preventing the initiation of necroptosis, the termini would need to be in complex for enough time to prevent RIPK1 binding which is inconsistent with our data.

Our memFRET approach is also potentially impacted by non-specific fluorescence and incomplete labelling of the channel. However, it has other potential concerns as well. (1) The C18-NTA could bind to the CTD in addition to intercalating into the membrane. However, the quenching is long lasting, and the measurements are made 5–20 min after exposure to the lipid which would mean this non-specific interaction would have to be quite durable. (2) The plasma membrane may deform around the channel putting the metal ions closer than expected to the L-ANAP in the CTD. (3) Similarly, the lipid could bind to the channel in the membrane artificially concentrating the metal ions around the channel. A number of lipids have been shown to bind and regulate ASICs, although ones with shorter acyl tails and no double bonds tend to show no effect on ASICs (*Klipp and Bankston, 2022*). (4) The distances measured between the intracellular domains and the membrane were based on a model of FRET with multiple acceptors in two dimensions (*Fung and Stryer, 1978*; *Gordon et al., 2016*). The model assumes a random arrangement of acceptors in the membrane and an approximate metal density of 0.002 molecules/ $\text{Å}^2$. We confirmed this approximate density using the approach detailed previously (*Gordon et al., 2016*). However, the previous use of this approach found FRET efficiencies higher than would have been predicted based on cryoEM structures. It is possible that the model underestimates the amount of metal in the membrane or the effect that multiple acceptors has on the fluorescence signal of the donor. (5) Finally, the most likely source of error comes from the highly nonlinear nature of FRET. As noted above, the closest approach distance is more highly represented in the overall FRET signal than other positions. The dynamic nature of these domains might be such that these intracellular domains spend some time in close proximity to the plasma membrane and it is these positions that dominate our signal.

We would argue that these limitations may impact the precise measurements made here but are unlikely to impact the overall trends. For instance, the dynamic nature of the NTD might mean we overestimate the proximity of the domain to the plasma membrane but is less likely to impact the conclusion that the proximal portion of this region spends a portion of its time mostly parallel to the membrane surface. Similarly, the distances we report for the position of the CTD relative to the membrane may be closer than reality, but the observation the acidification moves the domain away from the membrane is less likely impacted. Finally, the major goal of this project was to test the hypothesis that the NTD and CTD interact at rest and unbind during acidification and none of these caveats to our approach seem to suggest that our conclusion should be anything other than the N- and C-termini do not bind at resting pH.

## Conclusion

We believe the simplest interpretation of our results is that the NTD and CTD of ASIC1a do not interact at rest and that some other factor must be preventing RIPK1 binding under resting conditions. In addition, these data are most consistent with a model where the intracellular domains both spend significant time in close proximity to the plasma membrane with the CTD perhaps moving away during acidification. There is precedence for intracellular domains laying largely parallel to the plasma membrane. AMPA receptors are thought to have C-termini that are largely parallel to the membrane surface (*Zachariassen et al., 2016*) and the NTD of P2X receptors appears to lay along the plasma membrane across the pore region and form a 'cap' on the permeation pathway which can impact selectivity (*Gonzales et al., 2009*; *Tam et al., 2023*; *Yoder and Gouaux, 2020*). Given the structural relationship between P2X channels and ASICs and the results we show here, it will be interesting to see if perhaps the NTD of ASIC could act in a similar way.

Together our data suggest an alternative hypothesis for the inhibition of RIPK1 binding at rest. It is possible that the membrane itself acts to block RIPK1 binding to the CTD and that the change in conformation during acidification exposes the binding site by moving it away from the bilayer. This idea is consistent with the original findings for how ASIC1a triggers cell death in a RIPK1-dependent manner. Ultimately, more work needs to be done to understand this model of ASIC1a as a cell death

receptor and to further understand how acidosis controls the binding of RIPK1 to ASIC1a during ischemia.

# Methods

## Key resources table

| Reagent type (species) or resource | Designation | Source or reference | Identifiers | Additional information |
|---|---|---|---|---|
| Cell line (*Cricetulus griseus*) | CHO-K1 | ATCC | ATCC CCL-61 RRID:CVCL_0214 | |
| Recombinant DNA reagent | pANAP | Addgene: DOI: 10.1021/ja4059553 | Addgene: 48696 | |
| Recombinant DNA reagent | peRF1-E55D | Jason Chin Laboratory, MRC Laboratory of Molecular Biology, DOI: 10.1021/ja5069728 | | |
| Recombinant DNA reagent | rASIC1acDPCitrine.pCMV | Twist Bioscience | | |
| Recombinant DNA reagent | rASIC1acDPCitrine.pCMV TAG and Cysteine mutants | This paper | | Primers available upon request |
| Antibody | Purified Rabbit Anti-GFP, polyclonal | Torrey Pines Biolabs | Torrey Pines Biolabs Cat# TP401 071519, RRID:AB_10013661 | Western blots (1:5000) |
| Antibody | IRDye 680Rd Goat anti-Rabbit IgG, polyclonal | LI-COR Biosciences | LI-COR Biosciences Cat# 926–68071, RRID:AB_10956166 | Western blots (1:15,000) |
| Chemical Compound | 1-(2-pyridin-2-yldisulfanyl)ethyl)–1,4,7,10-tetraazacyclododecane (TETAC) | Toronto Research Chemicals | Toronto Research Chemicals Cat#P991915 | |
| Chemical Compound | N2,N2-bis(carboxymethyl)-N6-(1-oxooctadecyl)-l-lysine, C18-NTA | Toronto Research Chemicals | Toronto Research Chemicals Cat#S686540 | |
| Chemical Compound | L-ANAP-OMe | AsisChem Inc. | AsisChem Inc Cat#0146 | |
| Peptide | rASIC1a C-terminus | LifeTein | | |
| Software, Algorithm | pClamp and clampfit | Molecular Devices | RRID:SCR_011323 | |
| Software, Algorithm | Graphpad Prism 8 | Graphpad | RRID:SCR_002798 | |
| Software, Algorithm | ImageJ | NIH DOI: https://doi.org/10.1038/nmeth.2089 | RRID:SCR_003070 | |

## Molecular biology- constructs

A CMV vector containing ASIC1a from rat with conservative mutations listed in *Figure 1—figure supplement 1*, a short proline rich linker based on a flexible region from DNA polymerase, previously shown to have minimal effect of ASIC3 gating (*Klipp et al., 2020*), followed by an mCitrine tag on the C-terminus was synthesized (Twist Bioscience; San Francisco, CA). TAG and cysteine mutations were introduced using oligonucleotide-based mutagenesis with KOD Hot Start Master Mix according to manufacturer's instructions (Merk Millipore; Burlington, MA). All sequences were confirmed with DNA sequencing (ACGT DNA Sequencing Services; Wheeling, IL). pANAP was a gift from Peter Schultz (Addgene plasmid # 48696; http://n2t.net/addgene:48696; RRID:Addgene_48696; *Chatterjee et al., 2013*). peRF1-E55D was a kind gift from the Chin Laboratory (MRC Laboratory of Molecular Biology, Cambridge UK).

## Cell culture and transfection

CHOK-1 cells were obtained from ATCC (CCL-61 lot # 70029706). Chinese Hamster Ovary cells were purchased from ATCC and used within 2 years of purchase date. Cells were frozen at passage 2 and thawed and used until passage 14. Cell morphology, growth rate, and transfection efficiency, and growth rate were monitored to ensure that the cell lines did not change over time. Cells tested negative

for mycoplasm contamination. Cells were cultured in F-12 Ham's Media with 1 mM Glutamine (Gibco; New York) supplemented with 10% Fetal Bovine Serum (FBS) and 1% Antibiotic-Antimycotic (Gibco) at 37 C in a humidified atmosphere of 5% Carbon dioxide. Media was replaced every 2–3 days. Cells were transfected by using the nucleofector (Lonza; Basel). Prior to detachment cells were washed with Dulbecco's Phosphate Buffered Saline (DPBS, Gibco; New York) without calcium and magnesium. Cells were detached with TrypLE Express, 1.5 mL for a T75 flask for 5 min at 37 ° C and resuspended in 80 µL of SF cell line solution (Lonza; Basel) per flask. Twenty µL of cell solution was transferred to Eppendorf tube containing DNA (1.7 ug of rASIC-TAG, 1.7 µg pANAP and 1.7 µg eRF) to mix and immediately transferred to 20 µL Nucleovette strip for transfection using CHO-K1 protocol, DT-133, on the 4D-Nucleofector. We settled on a 1:1:1 ratio of channel: pANAP: eRF with a total of 5 µg DNA as the best ratio and total amount of DNA to get robust channel expression with the least readthrough- data not shown. eRF is critical for high yield of L-ANAP expressing full-length channels (*Schmied et al., 2014*). Cells were immediately transferred to prewarmed media and then split into 3–4 wells containing 25 mm poly-lysine treated coverslips and 10 µM ANAP. Cells were cultured at 30 ° C/ 5% $CO_2$ overnight which has been shown to increase protein production and arrest cell proliferation (*Kaufmann et al., 1999*). Cells were imaged 24–48 hr after transection. Six-well plates were wrapped in aluminum foil to protect from light. Cell passage number never exceeded P15 to ensure robust transient expression.

## Chemicals

L-ANAP-Me (AsisChem Inc; Waltham, MA) stocks were made by dissolving in DMSO. These stocks were wrapped in aluminum foil and stored at 3 mM at – 20 °C. Individual aliquots were frequently freshly thawed and freeze/thaw cycles were avoided. For use, 10 µL of L-ANAP stock was added to 3 mL of media which was then added to the cells.

## Western blot

CHO-K1 cells were transfected with rASIC-TAG, pANAP and eRF and supplemented with 10 µM L-ANAP. Transfection was performed in a similar fashion with only one minor difference. The larger 100 µL nuclecuvette was used and a single 10 cm dish of cells was resuspended in 100 µL of Ingenio electroporation solution for nucleofection (Mirus; Madison, WI). Approximately 18 hr after transfection, cells were harvested for western blot analysis. Media was aspirated and cells were washed with ice-cold DPBS. Cells were removed using a cell scraper and collected by centrifugation at 1000 xg for 5 min. Protein concentration was first normalized using cell pellet weights and resuspended using 25 µL of lysis buffer (300 mM NaCl, 20 mM TEA, 2 mM EDTA, 20% glycerol, 1% DDM) per 10 mg cells. DDM and Halt Protease inhibitor (Thermo Fisher Scientific; Waltham, MA) were added at time of use. Cells were lysed via end-over-end mixing for 1 hr at 4 °C followed by a 45 min centrifugation at 17,000 x*g*. Cleared lysates were loaded by normalizing to total protein concentration using $A_{280}$. Samples were run on either 4–12% Bis-Tris precast gels (Thermo Fisher Scientific; Waltham, MA) or Novex WedgeWell precast 4–12% Tris-Glycine gels (Thermo Fisher Scientific; Waltham, MA) at 200 V for 30–45 min. Protein was transferred to a PDVF membrane using Trans-Blot Turbo Transfer System (Bio-Rad; Hercules, CA) at 7 V, 1.3 A for 7 min. Membrane was incubated in blocking buffer (TBS-T, 5% milk) at room temperature for 1 h. Membrane was then incubated overnight in TBS-T (20 Tris, 137 NaCl, 0.1% Tween20) supplemented with 1:5000 primary antibody, purified rabbit anti-GFP TP401, (Torrey Pines Biolabs; Houston, TX). Membrane was washed 5x5 min with TBS-T followed by incubation with 1:15,000 secondary antibody for 1 hr at RT while keeping protected from light, IRDye 680RD Goat anti-Rabbit 925–681817 (Licor; Lincoln, NE). Membrane was washed 5x5 min with TBS-T and imaged using Licor Odessey FC.

## Electrophysiology

Experiments were performed in the whole-cell patch clamp configuration 18 hr after nucleofection. Borosilicate glass pipettes (Harvard Apparatus) were pulled to a resistance of 3–7 MΩ (P-1000; Sutter Instrument) and filled with an internal solution containing (in mM) 20 EGTA, 10 HEPES, 50 CsCl, 10 NaCl, and 60 CsF, pH 7.2. Extracellular solution contained (in mM) 110 NaCl, 5 KCl, 40 NMDG, 10 MES, 10 HEPES, 5 glucose, 10 Trizma base, 2 $CaCl_2$, and 1 $MgCl_2$, and pH was adjusted as desired with HCl or NaOH. An Axopatch 200B amplifier and pCLAMP 10.7 (Axon Instruments) were used to

record whole-cell currents. Solution changes were achieved through rapid perfusion using a SF-77B Fast-Step perfusion system (Warner Instruments) and Fluorescence was visualized on an Olympus IX73 microscope with a CoolLED pE-4000 illumination system. Recordings were performed at a holding potential of −80 mV with a 5 kHz low-pass filter and sampling at 10 kHz.

To measure the pH dependence of activation, rapid solution changes from pH 8 to seven separate test pH values were performed on each cell. Test pH values were applied for 1.5 s, typically followed by 35 s at resting pH of 8. Longer holding times (up to 90 s) at resting pH were necessary in some cases to ensure minimal tachyphylaxis which was controlled by monitoring current amplitudes for each individual cell for at least two full recordings. Test pH currents were normalized to the maximally activating condition (pH 5.5) for each recording. Normalized data from a minimum of three separate cells for each construct was averaged and SEM was calculated to generate pH plots. Half- maximally activating pH values, $pH_{0.5}$, were calculated by fitting to a Hill-Type equation:

$$I = \frac{1}{1 + 10^{\left[(pH0.5-pHx)n\right]}}$$

(1)

Where n=Hill number.

## Microscale thermophoresis

Peptides were synthesized by LifeTein (Somerset, NJ) and resuspended according to manufacturer instructions and stored in individual aliquots at –80° C. rASIC-CT was titrated to 5 nM rASIC-NT647 using a 2 x dilution series between 3 mM and 4.8 µM in 50 mM Tris (pH 7.4), 150 mM NaCl, 0.005% Tween in a total volume of 16 µL. After 5 min incubating at room temperature, the mixtures were transferred to Monolith NT.115 series premium coated capillaries. MST using a Monolith NT.115Pico instrument was performed using Pico-RED channel, 10% excitation power, medium MST power at room temperature (NanoTemper Technologies; Munich Germany).

> rASIC-CT
> KHRLSRRGKSQKEAKRSSADKGVALSLDDVKRHNPSESLRGHPAGMTYAANILPHHPARGTFEDFTS
> rASIC-NT647
> MELKTEEEEVGGVQPVSC(Alexa 647)

## Cell unroofing and imaging

Coverslips were washed 5 times with DPBS before adding 1 mL swell buffer (1 part Stabilization Buffer Tris (SBT) pH 8 : 3 parts water) for 30 s-2 min. Swell buffer was removed and cells were subjected to a mechanical force of liquid using 1.5 mL SBT pH 8 (15 mM Tris, 15 mM MES, 70 mM KCl, 1 mM MgCl) in a 5 mL syringe with a 22Gx1.5 inch PrecisionGlide needle attached (Becton, Dickinson; Franklin Lakes, NJ). Coverslip was subjected to a gentle steady stream of buffer, with the needle 1–2 inches away. Buffer was removed and cells were washed 5 x with SBT pH 8 or pH 6.

tmFRET experiments with unroofed cells were performed using an Olympus IX83 inverted microscope and a UPlanSAPo 60 x/1.35 oil immersion objective (Olympus; Shinjuku, Japan). Illumination was provided by a SOLA SE 5-LCR-VB light engine (Lumencor; Beaverton, OR). The filters used for L-ANAP excitation and emission were 357/44 nm and 460/80 nm, respectively. The filters used for mCitrine excitation and emission were 509/22 nm and 544/24 nm, respectively. Images were collected using a 500ms, 20% power L-ANAP exposure and 100ms, 5% power mCitrine exposure using an ORCA Flash4.0Lt CMOS Camera (Hamamatsu; Shizuoka, Japan).

25 mm coverslips that had already been unroofed were mounted onto Quick Exchange Platform with a RC-40LP chamber (Warner; Hollistan MA) on the microscope stage. All solutions were applied directly to the chamber with a pipette. 1 mL x 5 was added for all washes to ensure complete washout of the chamber. TETAC, C18, DTT and EDTA were added in the amounts noted in the following sections.

## Labeling with C18-NTA

C18-NTA (N2,N2-bis(carboxymethyl)-N6-(1-oxooctadecyl)-l-lysine) was made into individual 10 mM stocks in DMSO and stored at –20 °C. C18-NTA was diluted to 10 µM in Stabilization Buffer Tris (SBT)

for tmFRET experiments. Individual aliquots were freshly thawed for each experiment and freeze/thaw cycles were avoided.

A typical memFRET experiment followed these steps. Cells were unroofed as described in unroofing section. Coverslip was searched using mCitrine to look for unroofed cells expressing rASIC-TAG and these cells were marked using the Stage Navigator position finder in cellSens software (Olympus; Shinjuku, Japan). Initial images were taken at either pH 8 or pH 6. Solution is removed and 1 mL of 10 μM C18-NTA is added for 5 min x2. C18-NTA was removed, and coverslip was washed 5 x with SBT. 1 mL of 11 μM CoSO$_4$ was added for 1 min x2. Coverslip was washed 5 x with SBT at intended pH prior to taking the FRET image to remove any unbound or loosely bounded cobalt. FRET image was taken in appropriate pH SBT. One mL 10 mM EDTA was added for 1–2 min and removed. Coverslip was washed 5 x with SBT at intended pH and the final image was taken in appropriate pH SBT.

## Labeling with Cu²⁺-TETAC

TETAC was made into individual 100 mM stocks in DMSO and stored at –20 °C. For tmFRET experiments, 1 μL TETAC stock and 1 μL of 110 mM CuSO$_4$ were mixed together and incubated for 1 min. The solution turned a deep blue, indicating that the Cu²⁺ was bound to the TETAC cyclen ring. This mixture was diluted to 1 mM TETAC and 1.1 mM CuSO$_4$ by adding 98 μL of SBT. This solution was then brought up to 10 mL so that the Cu²⁺-TETAC was used at a final concentration of 10 μM. There is a 10% over-abundance of Cu²⁺ to ensure saturation of the TETAC. Individual aliquots were freshly thawed for each experiment and freeze/thaw cycles were avoided.

A typical ACCuRET experiment followed the ensuing steps. Cells were unroofed as described in unroofing section. Coverslip was searched using mCitrine to look for unroofed cells expressing rASIC-TAG and these cells were marked using the Stage Navigator position finder in cellSens software (Olympus; Shinjuku, Japan). Initial image was taken with either pH 8 or pH 6 SBT. Solution is removed and 1 mL of pre-reacted 10 μM Cu²⁺-TETAC was applied for 1 min x2. Coverslip was washed 5 x with SBT at intended pH prior to taking the FRET image to remove any unreacted Cu²⁺-TETAC. FRET image was taken in intended pH SBT. 1 mL 1 mM DTT was added for 1–2 min and removed. Coverslip was washed 5 x with SBT at intended pH and the final image was taken in intended pH SBT.

## Image analysis

Images were analyzed using ImageJ (National Institutes of Health; Bethesda, MD; *Schneider et al., 2012*). Regions of interest (ROI) were selected by thresholding the image based on the mCitrine channel and using the wand tool to automatically select the cell perimeter. Regions of the cell that were not unroofed were omitted from the selected ROI. For each cell, a nearby background region containing no cell or debris was selected. The mean gray value of the background ROI was subtracted from the mean gray value of the ROI for the corresponding cell. Photobleaching was measured by performing mock experiments at multiple TAG positions where no metal was added but washes and time between imaged remained the same at both pH 8 and pH 6. L-ANAP fluorescence was normalized to the first image. The normalized L-ANAP fluorescence of the second images, which represents the FRET image, were averaged together across multiple images and multiple experiments. We report our FRET as FRET efficiency (E) and correct the ANAP fluorescence for bleaching at pH 8 and pH 6 using the following equation:

$$E = 1 - \frac{F_{FRET}}{F_{bleach}} \qquad (2)$$

We confirmed there were no significant additional sources of energy transfer (mCitrine quenching, solution quenching etc) by performing the tmFRET experiments with zinc rather than copper and cobalt. Zinc has been shown to be coordinated by TETAC and NTA and does not measurably absorb light (*Lacerda et al., 2007*). We chose to not adjust the data to account for this nonspecific quenching because the measurements showed there was no notable additional quenching, however, do include these measurements on our FRET graphs (*Figures 6A and 8D*).

## Spectrophotometry/fluorometry measurements

Absorption measurements for Cu²⁺, Co²⁺, Zn²⁺ free in solution, as well as bound to either TETAC or C18-NTA were made using a NanoDrop One$^c$ (ThermoFisher Scientific; Waltham, MA). The absorption

spectra of 2 mM Cu²⁺-TETAC and 2 mM Co²⁺-C18-NTA were measured in both pH 6 and pH 8 SBT. Measurements were repeated at least three times and the data were averaged together. The extinction coefficient was calculated using the averaged absorbances.

In order to most closely mimic L-ANAP being incorporated into the protein backbone, we reacted L-ANAP with amine reactive agarose beads according to manufacturer's protocol. (ThermoFisher). We then measured L-ANAP spectra at pH 8 and pH 6 using an IsoPlane 160 (Princeton Instruments, Trenton, NJ) attached to our same microscope configuration from our tmFRET experiments.

### Distance calculations for ACCuRET

The $R_0$ for L-ANAP with Cu²⁺-TETAC was calculated using the following equation. The $R_0$ describes the distance that predicts 50% energy transfer between the donor and acceptor. We calculate two different $R_0$ values for our two pH conditions (Cu²⁺-TETAC pH 8, Cu²⁺-TETAC pH 6):

$$R_0 = C \sqrt[6]{\left( J Q \eta^{-4} \kappa^2 \right)}$$

(3)

$C$ is a scaling factor, $J$ is the normalized spectral overlap of ANAP emission at pH 8 or pH 6 with either Cu²⁺-TETAC pH 8 or Cu²⁺-TETAC pH 6, $Q$ is the quantum yield of L-ANAP, $n$ is the index of refraction, which is 1.33 here, and $\kappa^2$ is the orientation factor which we assumed to be 2/3 based on the assumption of dynamic isotropic orientation. For $Q$, we use 0.22 based on previous determination of L-ANAP quantum yield in SBT (*Zagotta et al., 2016*). Distances for each TAG position relative to the plasma membrane or to the singular cysteine in the CTD was calculated using the Förster equation:

$$r = R_0 \sqrt[6]{\frac{1}{E} - 1}$$

(4)

### Distance calculations for memFRET

To determine the distances for our memFRET approach, we needed to determine the relationship between distance and FRET efficiency using the theoretical dependence of FRET efficiency on closes approach distance. To do this, we used a previously published approach (*Gordon et al., 2016*) which was based on Förster's distance-dependent model of energy transfer (*Förster, 1949*) as described for the FRET pair rhodamine B and Co²⁺. To do this, we calculated the $R_0$ for ANAP and Co²⁺-C18-NTA (See below for values) again using 0.22 for the quantum yield, 2/3 for $\kappa$, and a refraction index of 1.33. The density of Co²⁺-C18-NTA in the membrane was assumed to be ~0.002 molecules/ Å². This assumption came from two sources. First, this value was previously determined by measuring the quenching of rhodamine B in the membrane by the same Co²⁺-C18-NTA (*Gordon et al., 2016*). We confirmed this result by repeating the same experiment and arrived at the same density of Co²⁺-C18-NTA in the membrane (data not shown).

$R_0$ values calculated for ACCuRET and memFRET were:

### Statistics

Data for tmFRET experiments are expressed as mean ± SEM of n independent unroofed cells. Statistical tests were performed in GraphPad Prism version 9.3.1 for Windows (GraphPad Software; San Diego, CA). Significance was determined using two-way ANOVA with a Tukey's post-hoc test for multiple comparisons unless otherwise noted. Significance levels: ns indicates $p > 0.05$, * indicates $p \leq 0.05$, ** indicates $p \leq 0.01$, *** indicates $p \leq 0.001$, **** indicates $p \leq 0.0001$. p-Values are all reported in the Source Data files associated with each figure. All replicates in this study are biological replicates. All data generated or analyzed are included in the figures.

## Acknowledgements

This work was supported by the National Eye Institute R00 EY024267 (to JR Bankston) and the National Institute of General Medical Sciences R35 GM137912 (to JR Bankston), National Institute of Dental and Craniofacial Research F31 DE028739 (to MM Cullinan) as well as funding from National Heart, Lung, and Blood Institute T32 HL007822 (to RC Klipp).

## Additional information

### Funding

| Funder | Grant reference number | Author |
|---|---|---|
| National Institute of General Medical Sciences | GM137912 | John R Bankston |
| National Eye Institute | EY024267 | John R Bankston |
| National Institute of Dental and Craniofacial Research | DE028739 | Megan M Cullinan |
| National Heart, Lung, and Blood Institute | HL007822 | Robert C Klipp |

The funders had no role in study design, data collection and interpretation, or the decision to submit the work for publication.

### Author contributions

Megan M Cullinan, Conceptualization, Data curation, Formal analysis, Funding acquisition, Investigation, Visualization, Writing - original draft, Writing - review and editing; Robert C Klipp, Data curation, Formal analysis, Investigation, Visualization; Abigail Camenisch, Investigation; John R Bankston, Conceptualization, Data curation, Supervision, Funding acquisition, Visualization, Writing - original draft, Project administration, Writing - review and editing

### Author ORCIDs

Megan M Cullinan (ID) http://orcid.org/0000-0001-7025-8507
John R Bankston (ID) http://orcid.org/0000-0002-9478-2335

Reviewer #1 (Public Review): https://doi.org/10.7554/eLife.90755.3.sa1
Reviewer #2 (Public Review): https://doi.org/10.7554/eLife.90755.3.sa2
Reviewer #3 (Public Review): https://doi.org/10.7554/eLife.90755.3.sa3
Author Response https://doi.org/10.7554/eLife.90755.3.sa4

## Additional files

### Supplementary files

• MDAR checklist

### Data availability

All data and replicates used in the study are included in the respective figures. For any data where the individual replicates are not visible, source data is provided.

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
