## [Editor Report · eLife assessment]

This **valuable** study illuminates molecular movements of acid-sensing ion channels by combining advanced chemical biology and biophysical techniques. The evidence for the main claim, lack of interaction of molecular termini, is **compelling** and challenges prior models. This work is expected to pique interest in the ion channel signaling field, providing a fresh perspective.

---

## [Referee Report · Reviewer #1 (Public Review)]

Cullinan et al. explore the hypothesis that the cytoplasmic N- and C-termini of ASIC1a, not resolved in x-ray or cryo-EM structures, form a dynamic complex that breaks apart at low pH, exposing a C-terminal binding site for RIPK1, a regulator of necrotic cell death. They expressed channels tagged at their N- and C-termini with the fluorescent, non-canonical amino acid ANAP in CHO cells using amber stop-codon suppression. Interaction between the termini was assessed by FRET between ANAP and colored transition metal ions bound either to a cysteine reactive chelator attached to the channel (TETAC) or metal-chelating lipids (C18-NTA). A key advantage to using metal ions is that they are very poor FRET acceptors, i.e. they must be very close to the donor for FRET to occur. This is ideal for measuring small distances/changes in distance on the scales expected from the initial hypothesis. In order to apply chelated metal ions, CHO cells were mechanically unroofed, providing access to the inner leaflet of the plasma membrane. At high pH, the N- and C- termini are close enough for FRET to be measured, but apparently too far apart to be explained by a direct binding interaction. At low pH, there was an apparent increase in FRET between the termini. FRET between ANAP on the N-and C-termini and metal ions bound to the plasma membrane suggests that both termini move away from the plasma membrane at low pH. The authors propose an alternative hypothesis whereby close association with the plasma membrane precludes RIPK1 biding to the C-terminus of ASIC1a.

The findings presented here are certainly valuable for the ion channel/signaling field and the technical approach only increases the significance of the work. The choice of techniques is appropriate for this study and the results are clear and high quality. Sufficient evidence is presented against the starting hypothesis. I have a few questions about certain controls and assumptions that I would like to see discussed more explicitly in the manuscript.

--As discussed by the authors, the C-terminal citrine could potentially disrupt the hypothesized interaction between the N- and C-termini.

--There is apparent read-through of some of the stop codons in the absence of ANAP, which could complicate interpretation of the experiments. The largest amount of read-through is for the E6TAG, L18TAG, and H515TAG constructs, which were not used for further experiments. However, some degree of read-through is evident from western blots for V10TAG, Q14TAG, L41TAG, and A44TAG as well.

Since the epitope used for western blots is on the C-terminus of the protein, the blots do not show the fraction of truncated protein. As discussed by the authors, N-terminally truncated constructs would be too small to assemble into channels. In constructs with the TAG codon towards the C-terminus, there is the potential for co-assembly of full-length and truncated subunits into trimers. Truncated subunits would not contribute directly to the fluorescence signal, but could potentially have allosteric effects on the position of the C-termini of full-length ANAP-tagged constructs in the context of a mixed channel.

---

## [Referee Report · Reviewer #2 (Public Review)]

Summary:

The authors use previously characterised FRET methods to measure distances between intracellular segments of ASIC and with the membrane. The distances are measured across different conditions and at multiple positions in a very complete study. The picture that emerges is that the N- and C-termini do not associate.

Strengths:

Good controls, good range of measurements, advanced, well-chosen and carefully performed FRET measurements. The paper is a technical triumph. Particularly, given the weak fluorescence of ANAP, the extent of measurements and the combination with TETAC is noteworthy.

The distance measurements are largely coherent and favour the interpretation that the N and C terminus are not close together as previously claimed.

Weaknesses:

One difficulty, which admittedly is hard to address, is that we do not have a positive control for what binding of something to either N- or C-terminus would look like (either in FRET or otherwise).

One limitation is unroofing. The concept of interaction with intracellular domains is being examined. But the authors use unroofing to measure the positions, fully disrupting the cytoplasm. Thus it is not excluded that the unroofing disrupts that interaction. But this limitation is discussed adequately in the text.

---

## [Referee Report · Reviewer #3 (Public Review)]

Summary: The manuscript by Cullinan et al., uses ANAP-tmFRET to test the hypothesis that the NTD and CTD form a complex at rest and to probe these domains for acid-induced conformational changes. They find convincing evidence that the NTD and CTD do not have a propensity to form a complex. They also report these domains are parallel to the membrane and that the NTD moves towards, and the CTD away, from the membrane upon acidification.

Strengths:

The major strength of the paper is the use of tmFRET, which excels at measuring short distances and is insensitive to orientation effects. The donor-acceptor pairs here are also great choices as they are minimally disruptive to the structure being studies.

Furthermore, they conduct these measurements over several positions with the N and C tails, both between the tails and to the membrane. Finally, to support their main point, MST is conducted to measure the association of recombinant N and C peptides, finding no evidence of association or complex formation.

Weaknesses:

While tmFRET is a strength, using ANAP as a donor requires the cells to be unroofed to eliminate background signal. This causes two problems. First, it removes any possible low affinity interacting proteins such as actinin (PMID 19028690). Second, the pH changes now occur to both 'extracellular' and 'intracellular' lipid planes. Thus, it is unclear if any conformational changes in the N and CTDs arise from desensitization of the receptor or protonation of specific amino acids in the N or CTDs or even protonation of certain phospholipid groups such as in phosphatidylserine. The authors do mention this caveat. But until a new approach is developed, the concerns over disruption by unroofing/washing and relevance of the changes remain.

Upon acidification, NTD position Q14 moves towards the plasma membrane (Figure 8B). Q14 also gets closer to C515 or doesn't change relative to 505 (Figures 7C and B) upon acidification. Yet position 505 moves away from the membrane (Figure 8D). It's unclear how the NTD moves closer to the membrane, and to the CTD but yet the CTD moves further from the membrane. Future experiments or approaches may refine this model.

---

## [Author Response]

The following is the authors’ response to the original reviews.

We thank the reviewers and editor for their thoughful and careful evaluation of our manuscript. We appreciate your time and effort and have incorporated many of these suggestions to improve our revised manuscript.

**Reviewer #1 (Public Review):**
Summary: Cullinan et al. explore the hypothesis that the cytoplasmic N- and C-termini of ASIC1a, not resolved in x-ray or cryo-EM structures, form a dynamic complex that breaks apart at low pH, exposing a C-terminal binding site for RIPK1, a regulator of necrotic cell death. They expressed channels tagged at their N- and C-termini with the fluorescent, non-canonical amino acid ANAP in CHO cells using amber stop-codon suppression. Interaction between the termini was assessed by FRET between ANAP and colored transition metal ions bound either to a cysteine reactive chelator attached to the channel (TETAC) or metal-chelating lipids (C18-NTA). A key advantage to using metal ions is that they are very poor FRET acceptors, i.e. they must be very close to the donor for FRET to occur. This is ideal for measuring small distances/changes in distance on the scales expected from the initial hypothesis. In order to apply chelated metal ions, CHO cells were mechanically unroofed, providing access to the inner leaflet of the plasma membrane. At high pH, the N- and C- termini are close enough for FRET to be measured, but apparently too far apart to be explained by a direct binding interaction. At low pH, there was an apparent increase in FRET between the termini. FRET between ANAP on the N-and Ctermini and metal ions bound to the plasma membrane suggests that both termini move away from the plasma membrane at low pH. The authors propose an alternative hypothesis whereby close association with the plasma membrane precludes RIPK1 binding to the C-terminus of ASIC1a.Strengths: The findings presented here are certainly valuable for the ion channel/signaling field and the technical approach only increases the significance of the work. The choice of techniques is appropriate for this study and the results are clear and high quality. Sufficient evidence is presented against the starting hypothesis.Weaknesses: I have a few questions about certain controls and assumptions that I would like to see discussed more explicitly in the manuscript.My biggest concern is with the C-terminal citrine tag. Might this prevent the hypothesized interaction between the N- and C-termini? What about the serine to cysteine mutations? The authors might consider a control experiment in channels lacking the C-terminal FP tag.

While it is certainly possible that the C-terminal citrine tag is preventing the hypothesized interaction between the intracellular termini, there are a few things that mitigate (but not eliminate) this concern. First, previous work looking at the interaction between the intracellular termini used FPs on both the N- and C-termini and concluded that in fact there is an interaction (PMID:31980622). Our channels have only a single FP, and we use a higher resolution FRET approach. Second, we aVach our citrine tag with a 11-residue linker, allowing for enhanced flexibility of the region and hopefully allowing for more space for an interaction that was posited to be between the very proximal part of the C-terminus (near the membrane and away from the tag) and the untagged N-terminus. Third, we previously showed that Stomatin, a much larger protein than the NTD, could bind the distal C-terminus of rASIC3 with a large fluorescent protein connected by the same linker on the C-terminus. In the case of Stomatin, the interaction involved the residues at the distal portion of the C-terminus close to the bulky FP. Interestingly, while we did not publish this, without this flexible linker, Stomatin could not regulate the channel and likely did not bind.

Despite this, we agree that this is possible and have added a statement in our limitations section explicitly saying this.

Figure 2 supplement 1 shows apparent read-through of the N-terminal stop codons. Given that most of the paper uses N-terminal ANAP tags, this figure should be moved out of the supplement. Do Nterminally truncated subunits form functional channels? Do the authors expect N-terminally truncated subunits to co-assemble in trimers with full-length subunits? The authors should include a more explicit discussion regarding the effect of truncated channels on their FRET signal in the case of such co-assembly.

The positions that show readthrough (E6, L18, H515) were not used in the study. We eliminated them largely on the basis of these westerns. We elected to put the bulk of the blots in the supplement simply because of how many there were. We believe this is the best compromise. It allows us to show representative blots for all our positions without making an illegible figure with 7 blots.

The N-terminally truncated subunits would create very short peptides that are not able to create functional channels. A premature stop at say E8 would create a 7-mer. Our longest N-terminal truncation would only create a protein of 32 amino acids. These don’t contain the transmembrane segments and thus cannot make functional channels.

As the epitope used for the western blots in Figure 2 and supplements is part of the C-terminal tag, these blots do not provide an estimate of the fraction of C-terminally truncated channels (those that failed to incorporate ANAP at the stop codon). What effect would C-terminally truncated channels have on the FRET signal if incorporated into trimers with full-length subunits?

Alternatively, C-terminally truncated subunits would be able to form functional channels because they contain the full N-terminus, the transmembrane domains, the extracellular domain and a portion of the C-terminus. We don’t think this is a major contaminant to our experiments. The only two C-terminal ANAP positions we use are 464 and 505. In each of these cases, they are only used for memFRET. The ones that do not contain ANAP are essentially “invisible” to the experiment. Since we are measuring their proximity to the membrane, having some missing should not maVer. However, there is some chance that truncations in some subunits could allosterically affect the position of the CT in other subunits. We have added a discussion of this in the manuscript.

Some general discussion of these results in the context of trimeric channels would be helpful. Is the putative interaction of the termini within or between subunits? Are the distances between subunits large enough to preclude FRET between donors on one subunit and acceptor ions bound on multiple subunits?

Thank you for this comment. We did not directly test whether the distances are within or between subunits. We considered using a concatemer to do this, however, the concatemeric channels do not express particularly well. Then, UAA incorporation hurts the expression as well. It was unlikely we would be able to get sufficient expression for tmFRET.

However, the Maclean group has previously tested this using FRET between concatenated subunits and determined that FRET is stronger within than between subunits. We have updated the manuscript to reflect a more thorough discussion of our results in the context of their trimeric assembly.

The authors conclude that the relatively small amount of FRET between the cytoplasmic termini suggests that the interaction previously modeled in Rosetta is unlikely. Is it possible that the proposed structure is correct, but labile? For example, could it be that the FRET signal is the time average of a state in which the termini directly interact (as in the Rosetta model) and one in which they do not?

The proposed RoseVa model does not include the reentrant loop of the channel, so it is probable that this model would change if it were redone to include this new feature of the channel.

However, we do discuss the limitation of FRET as a method that measures a time average that is weighted towards closest approach in our discussion section. The termini are most certainly dynamic and it is possible that spend some time in close proximity. Given that FRET is biased towards closest approach, we actually think this strengthens our argument that the termini don’t spend a great deal of time in complex. In addition, our MST data suggests that the termini do not bind. We have added some commentary on this to the discussion section for clarity.

**Reviewer #2 (Public Review):**
Summary:The authors use previously characterised FRET methods to measure distances between intracellular segments of ASIC and with the membrane. The distances are measured across different conditions and at multiple positions in a very complete study. The picture that emerges is that the N- and C-termini do not associate.Strengths:Good controls, good range of measurements, advanced, well-chosen and carefully performed FRET measurements. The paper is a technical triumph. Particularly, given the weak fluorescence of ANAP, the extent of measurements and the combination with TETAC is noteworthy.The distance measurements are largely coherent and favour the interpretation that the N and C terminus are not close together as previously claimed.Weaknesses:One difficulty is that we do not have a positive control for what binding of something to either N- or Cterminus would look like (either in FRET or otherwise).

We acknowledge that this is a challenge for the approach. Having a positive control for binding would be great but we are not sure such a thing exists. You could certainly imagine a complex between two domains where each label (ANAP and TETAC) are pointed away from one other (giving comparatively modest quenching) or one where they are very close (giving comparatively large quenching), both of which could still be bound. This is essentially a less significant version of the problem with using FPs to measure proximity…they are not very good proxies for the position of the termini. These small labels are certainly beVer proxies but still not perfect. Our conclusion here is based more on the totality of the data. We tried many combinations and saw no sign of distances closer than ~ 20A at resting pH. We think the simplest explanation is that they are not close to one another but we tried to lay out the limitations in the discussion.

One limitation that is not mentioned is the unroofing. The concept of interaction with intracellular domains is being examined. But the authors use unroofing to measure the positions, fully disrupting the cytoplasm. Thus it is not excluded that the unroofing disrupts that interaction. This should be mentioned as a possible (if unlikely) limitation.

Thank you for your comment. We discuss unroofing as a potential limitation because it exposes both sides of the plasma membrane to changes in pH. We have updated this section to include acknowledgement of the possibility that unroofing disrupts the interaction via washout of other critical proteins.

**Reviewer #3 (Public Review):**
Summary: The manuscript by Cullinan et al., uses ANAP-tmFRET to test the hypothesis that the NTD and CTD form a complex at rest and to probe these domains for acid-induced conformational changes. They find convincing evidence that the NTD and CTD do not have a propensity to form a complex. They also report these domains are parallel to the membrane and that the NTD moves towards, and the CTD away, from the membrane upon acidification.Strengths:The major strength of the paper is the use of tmFRET, which excels at measuring short distances and is insensitive to orientation effects. The donor-acceptor pairs here are also great choices as they are minimally disruptive to the structure being studied.Furthermore, they conduct these measurements over several positions with the N and C tails, both between the tails and to the membrane. Finally, to support their main point, MST is conducted to measure the association of recombinant N and C peptides, finding no evidence of association or complex formation.Weaknesses:While tmFRET is a strength, using ANAP as a donor requires the cells to be unroofed to eliminate background signal. This causes two problems. First, it removes any possible low affinity interacting proteins such as actinin (PMID 19028690). Second, the pH changes now occur to both 'extracellular' and 'intracellular' lipid planes. Thus, it is unclear if any conformational changes in the N and CTDs arise from desensitization of the receptor or protonation of specific amino acids in the N or CTDs or even protonation of certain phospholipid groups such as in phosphatidylserine. The authors do comment that prolonged extracellular acidification leads to intracellular acidification as well. But the concerns over disruption by unroofing/washing and relevance of the changes remain.

We acknowledge that unroofing is a limitation of our approach and noted it in the discussion. However, we have updated the section to include the possibility that the act of unroofing and washing could also disrupt the potential interaction between the intracellular domains as well as between these domains and other intracellular proteins. This was the best approach we could use to address our questions and it required that we unroof the cells. However, we look forward to future studies or new techniques that do not require the unroofing of the cells.

The distances calculated depend on the R0 between donor and acceptor. In turn, this depends on the donor's emission spectrum and quantum yield. The spectrum and yield of ANAP is very sensitive to local environment. It is a useful fluorophore for patch fluorometry for precisely this reason, and gating-induced conformational changes in the CTD have been reported just from changes in ANAP emission alone (PMID 29425514). Therefore, using a single R0 value for all positions (and both pHs at a single position) is inappropriate. The authors should either include this caveat and give some estimate of how big an impact changes spectrum and yield might have, or actually measure the emission spectra at all positions tested.

This is a reasonable concern and one we considered. Measuring the quantum yield would be quite difficult. However, we have measured spectra at a number of positions and see a relatively minimal shik in the peak. Most positions peak between 481 and 484nm. If you calculate the difference in R0 using theoretical spectra with a blue shik of 20nm, the difference in R0 is only ~1.5A. A shik of 20nm is on the higher side of anything we have seen in the literature (PMID 30038260) and since even with that large a shik, the difference is minimal we do not think measuring spectra for each position would impact the overall conclusions presented. As you noted, though, the quantum yield also changes. Assuming a change in yield from 0.22 to 0.47, the largest we found reported in the literature (PMID:29923827) , the R0 would increase by 2A. This same paper showed that the blue shiked position was the one with the higher extinction coefficient so these changes would be working in opposition to one another making the difference in R0 even smaller. It is important to note, that while tmFRET is a much more powerful measure of distance than standard FRET, these distances, as you point out, are quite challenging to measure precisely. Our conclusions are based less on the absolute distances and more on the observation that no positions show large quenching and that if there is any change upon acidification, it is in the wrong direction.

Overall, the writing and presentation of figures could be much improved with specific points mentioned in the recommendations for authors section.

See below.

The authors argue that the CTD is largely parallel to the plasma membrane, yet appear to base this conclusion on ANAP to membrane FRET of positions S464 and M505. Two positions is insufficient evidence to support such a claim. Some intermediate positions are needed.

We do not see in the paper where we suggest that the CTD is parallel. However, your point that we could try and determine if this was the case is correct. However, we aVempted to create several other CTD TAG mutants but struggled with readthrough and poor expression of these mutants so we opted to just include S464 and M505. Our point from these data is only that the distal CTD (505) must spend significant time near the membrane to explain our FRET data.

Upon acidification, NTD position Q14 moves towards the plasma membrane (Figure 8B). Q14 also gets closer to C515 or doesn't change relative to 505 (Figures 7C and B) upon acidification. Yet position 505 moves away from the membrane (Figure 8D). How can the NTD move closer to the membrane, and to the CTD but yet the CTD move further from the membrane? Some comment or clarification is needed.

This is a reasonable question and one that is hard to definitively answer. Our goal here was to test the hypothesis that the termini are bound at rest. Mapping the precise positions of the termini is difficult for reasons we will enumerate in the question that asks why we didn’t make a model. There are potentially multiple explanations but the easiest one would be that the CTD could move away from the membrane but closer to Q14, for instance, if the distal termini, say, rotated towards the NTD. This would move 505 closer and have no impact on whether or not the NTD and CTD moved away or toward the membrane.

**Reviewer #1 (Recommendations For The Authors):**
Minor concernsThe authors show the spectrum of ANAP attached to beads and use this spectrum to calculate R0 for their FRET measurements. Peak ANAP fluorescence is dependent on local environment and many reports show ANAP in protein blue-shiked relative to the values reported here. How would this affect the distance measurements reported?

This is an important point. See above for the answer.

Could the lack of interaction between the N- and C-terminal peptides in Figure 7 arise from the cysteine to serine mutations or lack of structure in the synthetic peptides. How were peptide concentrations measured/verified for the experiment?

It is possible that cysteine to serine mutations could prevent the interaction. It is also possible that these peptides are not capable of adopting their native fold without the presence of the plasma membrane or due to being synthetically created. However, the termini are thought to be largely unstructured. We received these peptides in lyophilized form at >95% purity and resuspended to our desired stock concentration (3 mM C-terminus, 1 mM N-terminus). Even if our concentration was off, we see no signs of interaction up to quite a high concentration.

How was photobleaching measured for correcting the data?

We executed several mock experiments at various TAG positions using either pH 8 and pH 6, where we performed the experiments as usual but with a mock solution exchange when we would normally add the metal. We normalized the L-ANAP fluorescence to the first image and averaged together these values for pH 8 and pH 6. We then corrected using Equation 2 in the manuscript..

We have updated the methods to include how we adjusted for bleaching.

The authors may wish to make it more explicit that their Zn2+ controls also preclude the possibility that a changing FRET signal between ANAP and citrine may affect their data.

Thank you for this comment. We agree, it would strengthen the manuscript to include this statement. We have now included this.

It might be useful to the reader if the authors could include (as a supplement) plots of their data (like in Figure 6), in which FRET efficiency has been converted to distance.

We considered this idea as well but felt like showing the actual data in the figures and the distances in a table would be best.

Figure 5D is mentioned in the text before any other figures. This is unconventional. Could this panel be moved to Figure 1 or the mention moved to later?

Changed

western blot is not capitalized.

Changed.

Figure 1, the ANAP structure shown is the methyl ester, which is presumably cleaved before ANAP is conjugated to the tRNA. The authors may wish to replace this with the free acid structure.

This is a fair point. We originally used the methyl ester structure to indicate the version of ANAP we chose to use. However, you are correct that the methyl ester is cleaved before conjugation to the tRNA. We replaced the methyl ester with the free acid structure to clarify this.

Figures 1 and 4 should have scale bars for the images.

Scale bars have been added to figures 1, 4, and 5.

In Figure 3, the letters in the structures (particularly TETAC) are way too small. Please increase the font size.

Changed

In Figure 3 and Figure 3 supplement 1, the axes are labeled "Absorbance (M-1cm-1)." Absorbance is dimensionless. The authors are likely reporting the extinction coefficient.

Thank you for catching this. We adjusted the axes to extinction coefficient.

In Figures 5 B and C, it might be clearer if the headers read "Initial, +Cu2+/TETAC, DTT" rather than "Initial, FRET, Recovery."

Changed

The panel labels for Figure 8 seem to be out of order.

Changed

The L for L-ANAP should be rendered, by convention, in small caps.

This is a good example of learning something new from the review process. This is the first I have ever heard of small caps. We can find no other papers that use small caps for L-ANAP so I am not 100% sure what convention this is referring to and don’t want to change the wrong thing in the paper. We are happy to change if the editorial staff at eLife agree but have lek this for now.

**Reviewer #2 (Recommendations For The Authors):**
With so many distances measured, why was not even a basic structural model attempted?

We certainly considered it, but a number of things lead us to conclude that it might imply more certainty about the structure of these termini than we hope to give. (1) Given that the FRET is a time average of positions, these distance constraints would not do much constraining. (2) Given that the termini are likely unstructured and flexible this makes the problem in 1 worse. (3) There is no structural information to use as a starting point for a model. (4) The flexibility of the linkers for each FRET pair also introduces uncertainty. This can, in theory, be modeled as they do in EPR but all of this together made us decide not to do this. What we hope readers take home, is the overall picture of the data is not consistent with the original RIPK1 hypothesis.

Maybe it would be good to draw a band on the graphs in Figure 6 for the FRET signal expected for interaction (and thus, disfavoured by these data). This would at least give context.

We agree this could be helpful, but it is not so easy to do. What distance would we choose? We could put a line at ~5Å (the model predicted distance). As we noted above, a number of distances could be compatible with an interaction. However, we think it’s unlikely that if a complex was formed that none of our measurements would show a distance closer than 20Å at rest and that an unbinding event would then lead to a decrease in distance. This, to us, is the take home message.

Minor points:"Aker unroofing the cells, only fluorescence associated with the "footprint", or dorsal surface, of the cell membrane is lek behind."The authors use dorsal and ventral in this section to describe parts of an adherent cell. But in the first instance, they remove the dorsal part of the cell, and then in this phrase, the dorsal part is lek behind....I am a bit confused.

Thank you for pointing out this mistake, we have fixed this. It is indeed the ventral surface lek behind.

"bind at rest an" - and?

Changed

"One previous study used a different approach to try and map the topography of the intracellular termini of ASIC1a comparable to our memFRET experiments." I think a citation is due.

Citation added

"great deal of precedent" even if this result is from my own lab, I would prefer that the authors note that it's one study from one lab! I think best just to delete "great deal of".

“Great deal of” deleted

I think the column "Significance" in the tables is unnecessary when the P value is given.

Thank you for this suggestion. We agree and have made the change.

Figure 7a Q14TAG has a clearly bimodal distribution at pH 8. What could be the meaning of this result? The authors do not mention it that I could find. Perhaps there is no meaning. The authors should state what they think is (or is not) going on.

This is a good question and we don’t have a good answer. It appears to be experimental variability. The data from the “low fret” in this experimental condition all came from the same days. So something was different that day. We considered that they might be outliers to exclude but thought showing all of our data was the beVer path. We reperformed the ANOVA here separating out the “outlier” day and nothing of substance changed. Both populations were still different with P value less than 0.001.

Typo: Lumencore

Changed

Maybe just a matter of taste but the panel created with Biorender in Figure 8 is not attractive and depicts the channel differently to in Figure 5D, which is again different from Figure 1A. Surely one advantage of using computer-generated artwork could be to have consistency.

We agree and have used the same cartoon for all of our images with the one exception being the schematics that are just meant to show the positions that are present in each bar graph.

Figure 4A was squashed to fit (text aspect ratio is wrong).

Fixed

**Reviewer #3 (Recommendations For The Authors):**
Citrine is used to report incorporation. Yet citrine has a strong tendency to dimerize (PMID 27240257). Did the authors use mCitrine or just Citrine? This is quite important in interpreting their data.

Thank you for pointing out this important distinction. We use mCitirine which we have added to the methods.

The manuscript has numerous instances of imprecise language. For example, page 10, last para, first line, "previous studies have looked at..." or page 7, final paragraph "tell a similar story". Related, the figures could be much better. For example, in Figure 1, where the authors depict the anap chemical in red, as opposed to the blue one might expect of a blue emiqng fluorophore. In figure 6, ANAP is also in red with the quenching group in green. This is opposite to how one typically thinks of FRET with the warmer color being the acceptor not the donor. Moreover, the pH 6 condition is also colored the same shade of red as the ANAP. Labels of Cys positions would again be useful here. In Figure 3, the heteroatoms of TETAC and C18-NTA are very small and difficult to see. It would also be good to label these structures, and the spectra below, so the reader can tell at a glance without looking at the caption, what the structures and spectra arise from. Also, how are the absorption spectra normalized? This is not discussed in the methods. The lack of attention to presentation mars an otherwise nice study.

Thank you for these points. We have made modifications to the manuscript to address these comments.

Abstract, second last line "Aker prolonged acidification, ...", 'prolonged' could be interpreted as 'it takes a while for the domain to move' or 'the movement only happens aker a while'. This not what the authors intend to convey. Consider modifying to just 'Aker acidification,'

We updated the main text to indicate that prolonged acidification is intended to describe acidification that occurs over the minutes timescale.

Pdf page 6, bottom para on Anap incorporation not altering channel function: What is meant by 'steady state pH dependence of activation'? This implies the authors applied a pH stimulus, then waited until equilibrium was achieved ie. until desensitization was complete and measured the current at that point. It seems more likely they simply applied different pH stimuli and measured the peak response and that the use of 'steady state' here is a typo.

We removed the phrase steady state.

Same section, controls of electrophysiology allude to 485, 505 and 515 ANAP-containing channels. In fact, the authors have no way of determining what fraction (if any) of the pH evoked currents arise from channels containing Anap in those positions versus from simply having a translation stop but still functioning. This should be mentioned.

This is correct. We cannot be sure the CTD TAG positions are not a mixture of ANAP-containing channels and truncations. See above for why we do not think this a big concern for the FRET experiments. Functionally, though, you are correct that we cannot tell. We now mention this in the paper.

Methods, the abbreviation for SBT should be defined somewhere.

Added.

Methods, unroofing section, middle paragraph, the authors use nM not nm to list wavelengths of light.

Changed.

Figure 3C-D: There's an unexpected blip in the Anap emission spectra at ~500 nm. Are the grating efficiency of the spectrograph and quantum efficiency of the camera accounted for in these spectra?

This is a good question. The data are not corrected for either camera efficiency or grating efficiency. We don’t have easy access to the actual data (although we can see a pdf version of each). There is a liVle blip in the grating efficiency graph that could partly explain the blip in our spectra.

Figure 5C, were recovery experiments routinely done? If so, would be good to show more than n = 1 in the plot to get an idea of reproducibility.

Recovery experiments were done in every experiment but are not shown for simplicity. We have included all FRET and recovery data for position Q14TAG-C469 at pH 6 in figure 5C to show reproducibility of our FRET and recovery data.

Table 1, considering adding a Δ distance column (pH 8 versus 6) so the magnitude of changes are more easily seen.

This is a reasonable suggestion but we decided not to include a Δ distance column. The data are whole numbers and people can easily determine the Δ distance. We felt that including that column would bring too much focus on what we think are preVy small changes. Our hope is that readers take away that the data are not consistent with complex formation between the determine and focus less on absolute distances.

Figure 7A, Q14tag pH 8 condition has a quite a bit of spread and, likely, two populations. These data, as well as G11, are unlikely to be parametric and hence ANOVA is inappropriate. A normality test, and likely Kruskal-Wallis test is called for.

Aker testing for normality, the data for Q14TAG C485 pH8 are non-normally distributed. However, a Kruskal Wallis is a non-parametric test for a one-way ANOVA and not applicable here. We separated the data out into population 1 and 2 and repeated the two-way ANOVA statistical test. When Q14TAG pH 8 is split into 2 populations, the statistics hardly change. When the data is not separated, Q14TAG pH 8 relative to pH 6 has a p-value <0.0001. When the 2 populations are separated, both populations relative to Q14TAG pH 6 still have a p-value of <0.0001.